# Integrating biomedical research and electronic health records to create knowledge-based biologically meaningful machine-readable embeddings

Charlotte A. Nelson [1], Atul J. Butte [2,3] & Sergio E. Baranzini [2,4]

In order to advance precision medicine, detailed clinical features ought to be described in a way that leverages current knowledge. Although data collected from biomedical research is expanding at an almost exponential rate, our ability to transform that information into patient care has not kept at pace. A major barrier preventing this transformation is that multi-dimensional data collection and analysis is usually carried out without much understanding of the underlying knowledge structure. Here, in an effort to bridge this gap, Electronic Health Records (EHRs) of individual patients are connected to a heterogeneous knowledge network called Scalable Precision Medicine Oriented Knowledge Engine (SPOKE). Then an unsupervised machine-learning algorithm creates Propagated SPOKE Entry Vectors (PSEVs) that encode the importance of each SPOKE node for any code in the EHRs. We argue that these results, alongside the natural integration of PSEVs into any EHR machine-learning platform, provide a key step toward precision medicine.

[1] Integrated Program in Quantitative Biology, University of California San Francisco, San Francisco, CA, USA. [2] Bakar Computational Health Sciences Institute, University of California San Francisco, San Francisco, CA, USA. [3] Department of Pediatrics, University of California San Francisco, San Francisco, CA, USA. [4] Weill Institute for Neuroscience. Department of Neurology, University of California San Francisco, San Francisco, CA, USA. Correspondence and requests for materials should be addressed to S.E.B. (email: Sergio.Baranzini@ucsf.edu)

The rate at which the ever growing body of world data is being transformed into information and knowledge in some areas (e.g., banking, e-commerce, etc.) far exceeds the pace of such process in the medical sciences. This problem is widely recognized as one of the limiting steps in realizing the paradigm of precision medicine, the application of all available knowledge to solve a medical problem in a single individual[1,2].

In order to address this issue, several efforts to integrate these data sources in a single platform are ongoing[3,4]. The basic premise of data integration is the discovery of new knowledge by virtue of facilitating the navigation from one concept to another, particularly if they do not belong to the same scientific discipline. One of the most promising approaches to this end makes use of heterogeneous networks. Heterogeneous networks are ensembles of connected entities with multiple types of nodes and edges; this particular disposition enables the merging of data from multiple sources, thus creating a continuous graph. The complex nature and interconnectedness of human diseases illustrates the importance of such networks[5]. Even bipartite networks, with only two types of nodes, have furthered our understanding on disease–gene relationships, and provided insight into the pathophysiological relationship across multiple diseases[6].

In an attempt to address one of the most critical challenges in precision medicine, a handful of recent studies has started to merge basic science-level data with phenotypic data encoded in electronic health records (EHRs) to get a deeper understanding of disease pathogenesis and their classification to enable rational and actionable medical decisions. One such project is the Electronic Medical Records and Genomics (eMERGE) Network. The eMERGE consortium collected both DNA and EHRs from patients at multiple sites. eMERGE and subsequent studies showed the advantages of using EHRs in genetic studies[7–9]. Another project-linked gene expression measurements and EHRs, an approach through which researchers were able to identify possible biomarkers for maturation and aging[10]. While these studies illustrate the benefits of combining data from basic science with EHRs, no efforts connecting EHR to a comprehensive knowledge network have been yet reported. This study builds upon these concepts and utilizes a heterogeneous network called Scalable Precision Medicine Oriented Knowledge Engine (SPOKE) to interpret data stored in EHRs of more than 800,000 individuals at The University of California, San Francisco (UCSF). SPOKE integrates data from 29 publicly available databases, such as the GWAS catalog, STARGEO, ChEMBL, LINCS, and GeneOntology, and contains over 47,000 nodes of 11 types and 2.25 million edges of 24 types, including disease–gene, drug–target, drug–disease, protein–protein, and drug–side effect (Supplementary Tables 1, 2)[11,12].

In this work, we describe a method for embedding clinical features from EHRs onto SPOKE. By connecting EHRs to SPOKE, we are providing real-world context to the network thus enabling the creation of biologically and medically meaningful "barcodes" (i.e., embeddings) for each medical variable that maps onto SPOKE. We show that these barcodes can be used to recover purposely hidden network relationships, such as *Disease–Gene*, *Disease–Disease*, *Compound–Gene*, and *Compound–Compound*. Furthermore, the correct inference of intentionally deleted edges connecting *SideEffect* to *Anatomy* nodes in SPOKE is also demonstrated.

## Results

**Embedding EHR concepts in a knowledge network**. The main strategy of this work is to embed EHRs onto the SPOKE knowledge network utilizing a modified version of PageRank, the well-established random walk algorithm[13]. These embeddings,

called Propagated SPOKE Entry Vectors (PSEVs), can be created for any group of subjects with a particular characteristic (i.e., patient cohort). Here, we describe the creation of PSEVs for patient cohorts selected using either discrete or continuous EHR variables. PSEVs are vectors in which each element corresponds to a node in SPOKE. Therefore, the length of each PSEV is equal to the number of nodes in SPOKE. Furthermore, the value of each element in a PSEV encodes the importance of its corresponding node in SPOKE for a given patient cohort.

De-identified structured EHR data from 816,504 patients were obtained from the UCSF Medical Center through UCSF Information Technology Services Academic Research Systems. These records were then filtered to only include patients that had been diagnosed with at least one of the 137 complex diseases currently represented in SPOKE, leaving 292,753 patients for further analysis. Select structured data tables from the EHR were used to identify EHR concepts that can be directly linked a node in SPOKE. These points of overlap between the EHRs and SPOKE are called SPOKE Entry Points (SEPs). The data tables were then used to create 3233 PSEVs, one for each identified SEP (see the Methods section). Each structured EHR table contains codes, referred to as EHR concepts, that can be linked to standardized medical terminology. EHR concepts can be diagnostic codes (ICD9CM or ICD10CM), medication order codes (translated to RxNorm), or lab codes (LOINC). Although 3233 represents a sizable proportion (7.5%) of all nodes in SPOKE, most nodes are not directly reachable, thus potentially diluting the power of the network's internal connectivity. To address this challenge, a modified version of the random walk algorithm was used to propagate all 3233 SEPs through the entirety of the knowledge network, thus creating a unique PSEV (i.e., medical profile) for each of the selected clinical features in the EHRs.

In the original random walk algorithm, a walker is placed onto a given node in a network, and it can move from one node to another as long as there is an edge connecting them. The algorithm was adjusted in a way similar to topic-sensitive PageRank[14], by weighing the restart parameter (1 - damping factor) of the random walker toward nodes that are important for a given patient population (cohort used for PSEV creation). Hence, the importance of a given SEP ($SEP_i$) is equivalent to the proportion of patients in the cohort that had an EHR concept in their records that mapped to $SEP_i$. This modified version of PageRank can be applied to any patient cohort.

**Benchmarking PSEVs with BMI**. To demonstrate that these vectors capture biologically meaningful information, PSEVs were created using body mass index ($BMI = weight \times height^{-2}$) (an ubiquitous variable in the EHR) as the basis to define cohorts. BMI is typically used to classify patients into four standard classes (underweight, normal, overweight, and obese). Decades of research have provided deep insight into both the phenotypic and mechanistic manifestation of obesity. However, only the top-level (phenotypic) information (i.e., BMI class) is captured in the EHRs. We hypothesized that by using this method it would be possible to integrate mechanistic and biological level data, thus gaining additional insight into the characteristics of people classified into each obesity class.

When examining the distribution of BMIs across the UCSF patient population, four groups are clearly distinguishable. These natural subpopulations were used to separate patients into four cohorts that aligned well with the standard BMI classes (Fig. 1a). Since the BMI thresholds only differed from that of the standard classes by −0.5 BMI, the four cohorts will be reference using the names of the corresponding standard classes (underweight, normal, overweight, and obese).

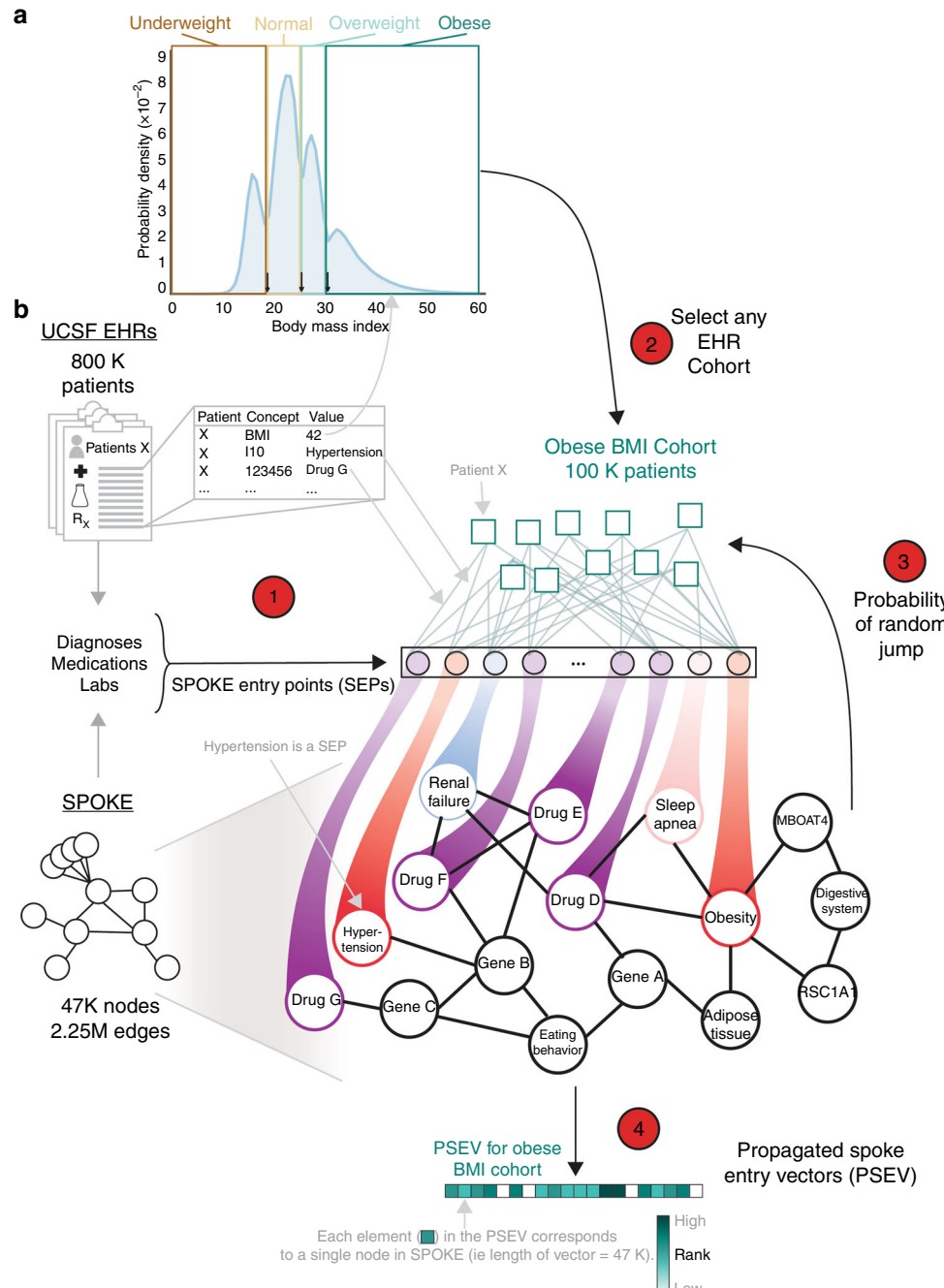

**Fig. 1** Embedding EHR concepts in a knowledge network. **a** Distribution of patient BMIs at UCSF. Four BMI cohorts were created using the natural boundaries of the BMI distribution (boxes I–IV: <18, 18–24.5, 24.6–29.5, and >29.6). Arrows at the bottom correspond to the BMIs that separate the standardize weight classes. **b** Step 1: find the overlapping concepts between SPOKE and the patient data (EHRs). These are called SPOKE Entry Points (SEPs). Step 2: choose any code or concept in the EHR to make cohort. Here, we have chosen patients with a high BMI (Cohort IV). Then connect each patient in the cohort to all of the SEPs in their records. Step 3: perform PageRank such that the walker restarts in the patient cohort. Iterate until desired threshold is reached. Step 4: final node ranks are then used to create the weights in the Propagated SPOKE Entry Vector (PSEV)

Figure 1b illustrates the modified PageRank algorithm using patients in the obese BMI cohort. First, the records from all 100,187 patients in the obese BMI cohort were extracted. Second, connections were created between each of those patients and all of their additional SEPs. By definition, this means connections to any medication, diagnosis, or laboratory result that was present in both that patient's record and SPOKE. Third, a random walker was initialized and allowed to either move to a neighboring node (optimized damping factor = 0.9) or randomly jump to any SEP with probability $\beta$ (optimized $\beta = 0.1$). However, $\beta$ was not evenly distributed among the SEPs (as in the original algorithm), but was instead weighted based on how important each SEP was for the cohort (Supplementary Fig. 1). This weight is akin to having the random walker jumping to a random patient in the cohort and traversing to one of that patient's SEPs (Supplementary Fig. 1A). Each iteration resulted in a rank vector that reflects the proportion of time the walker spent on each node in the network. In practice, for each iteration, this was calculated by taking the dot product of the transition probability matrix and the rank vector from the previous iteration (see the Methods section).

Once the algorithm converged, the rank vector from the final iteration was returned (bottom vector). This final rank vector is called a PSEV.

We imagine SPOKE as a set of interconnected water pipes and the SEPs as input valves. Then, the percentage of obese patients that also have type 2 diabetes in their EHRs will determine how much water is allowed to flow through the type 2 diabetes SEP valve (a measure of its importance). Once all of the valves have been calibrated to fit the obese patient population, water can then flow to downstream nodes in SPOKE. Once the water reaches an almost steady state, differential water flow will highlight intersections of pipes (SPOKE nodes) that are significant for obese patients. It should be noted that separating patients using other unbiased methods for separating patients or simply treating BMI as a continuous variable would lead to the same results (Supplementary Fig. 2). Therefore, a priori knowledge about a cohort is not necessary to create a meaningful PSEV.

**Identifying phenotypic traits in PSEVs.** The final PSEV is representative of how important each SPOKE node is for a given EHR concept based on both the connections in SPOKE and the patients with that concept in their EHR. Therefore, a PSEV was generated for each of the four BMI cohorts. Since each element in a PSEV corresponds to a single node in SPOKE, it is now possible to determine how important each SPOKE node is for each of the BMI cohorts. To examine the significance of this observations, values of the *Disease* elements in the PSEVs were compared for each of the four BMI cohorts. The top *Diseases* in the PSEV of the obese cohort were obesity, hypertension, type 2 diabetes mellitus, and metabolic syndrome X. While not unexpected, the identification of these diseases as the most important conditions for this group of patients, without any reference to the mechanisms underlying obesity present in the EHR, is noteworthy. These diseases also correlated well with average BMI (r = 0.75–0.95) and when their ranks were plotted against average BMI, they displayed some of the steepest slopes (slope = 5.4–6.7), suggesting they were causally related.

To learn more about the relationships between BMI and these potentially associated diseases, for each BMI class, we plotted the average BMI (mean BMI per cohort) against the rank of these *Disease* elements in the four respective PSEVs (Fig. 2a). The most noticeable differences in the rank were observed for the *Disease* element hypertension between the underweight and normal cohorts (rank increases from 136 to 17), and the *Disease* element obesity between overweight and obese (rank increases from 132 to 1). These makes sense given that hypertension was the most prevalent disease in UCSF cohort, and many of the factors that contribute to hypertension risk are also related to increasing BMI and the BMI classification of obesity is very similar to the threshold for our obese cohort. In addition, there was a major difference (111 positions) in the rank of the type 2 diabetes mellitus between the normal and overweight cohorts. This change suggests that type 2 diabetes mellitus became associated with BMI once patients have reached overweight status, and that an increased BMI was one phenotypic manifestation of this condition. Finally, metabolic syndrome X was highly ranked (position 4) only in the obese cohort. However, it differed from obesity in that the progression in rank between normal and overweight was gradual, suggesting increased BMI as a risk factor in metabolic syndrome X. In contrast, the rank of celiac and Crohn's disease progressively diminishes 116 and 108 positions, respectively, between the underweight and obese cohorts. This trend could be explained by the fact that weight loss is symptomatic of both celiac and Crohn's disease. Another *Disease* that progressively moves down in rank with increased BMI was

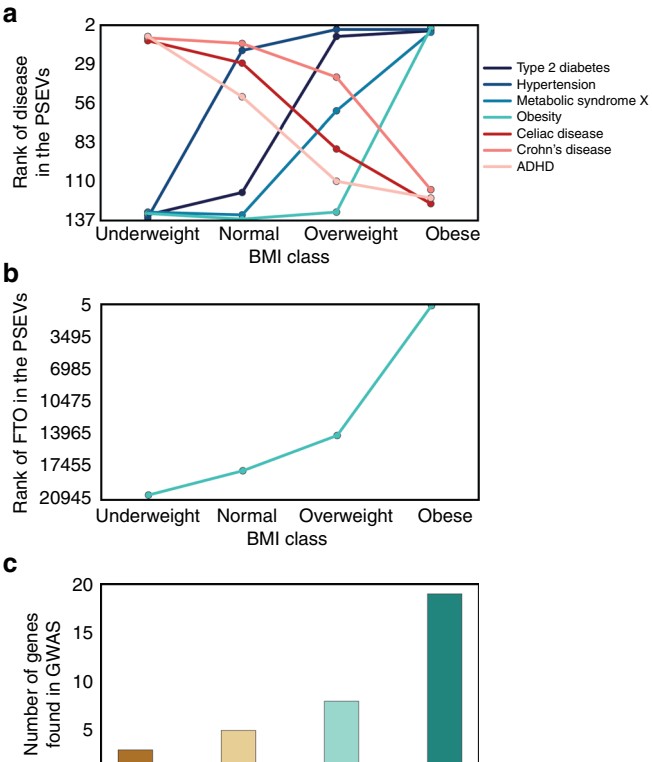

**Fig. 2** PSEVs contain phenotypic and genotypic information. **a** BMI cohort versus *Disease* rank. The top fourranked *Diseases* in the in cohort IV's PSEV were obesity, hypertension, type 2 diabetes mellitus, and metabolic syndrome X. All four show a positive relationship with BMI. The opposite trend was observed for celiac disease, Crohn's disease, and attention-deficit disorder which were highly ranked in cohort I's PSEV. **b** FTO gene was positively correlated with BMI. **c** The number of overlapping genes between the GWAS catalog for increased BMI (n = 365) and the top 365 *Genes* in each BMI cohort PSEV

attention-deficit hyperactivity disorder (ADHD). This negative correlation was due to the fact that most of the medications used to treat ADHD have side effects related to weight loss and loss of appetite. These results show that the algorithm correctly upweights phenotypes associated with high BMI in the PSEVs for the overweight and obese cohorts while also downweighting those phenotypes in the underweight and normal cohorts. Furthermore, these results were replicated treating BMI as a continuous variable instead of discrete classes (Supplementary Fig. 2). This replication shows the robustness of this approach, and is important given that most continuous variables in the EHRs are not associated with a fixed number of classes. It should be noted that up until this point, BMI has been treated as a continuous variable used to simply split patients into groups, and the algorithm has been blind to the standardized classes associated to those groups. BMI was chosen to illustrate the utility of PSEVs because the consequences/traits of an abnormal BMI are very well known. However, since a PSEV can be created for any variable in the EHRs, they can also be used to reveal phenotypic traits associated with less well-understood variables and phenotypes.

**PSEVs reveal genotypic traits and biological mechanisms.** To test whether the same trend was seen at the genotypic level, linear

regressions were computed on the average BMI versus *Gene* rank. Again, the genes that positively correlated with average BMI (mean BMI of a cohort) were given the top prioritization in the high BMI (obese) PSEV. An example of a gene that was positively correlated with BMI is alpha-ketoglutarate-dependent dioxygenase (FTO), also known as fat mass and obesity-associated protein, is shown in Fig. 2b. To check if these genes were genetically related to BMI, genes associated with increased BMI (not necessarily obesity, just an average increase) were extracted from the GWAS catalog ($n = 365$ mapped genes) and compared them to the top 365 ranked genes in the PSEVs. Remarkably, the PSEV for the obese cohort was significantly enriched in known BMI-associated genes ($p = 2.19E{-}10$, binomial test; Fig. 2c). The PSEV for the overweight cohort was also significant, while the BMI cohorts corresponding to underweight and normal BMIs showed no significant enrichment. Therefore, *Gene* elements that were highly ranked in the overweight or obese BMI PSEVs had a higher probability of harboring a susceptibility variant. These results illustrate that PSEVs can learn new biologically meaningful relationships.

**PSEVs preserve original SPOKE edges**. After identifying that the obese BMI PSEV was able to preserve the known gene expression edges in SPOKE, we decided to check this with other concepts in a high-throughput manner. To do this, we utilized the fact that SEPs are SPOKE nodes that can be directly mapped to EHR concept(s) to extract 3233 patient cohorts from the diagnosis, medication order, and lab tables in the EHRs. PSEVs were then generated for each of the cohorts. Then the top ranked nodes (ranked per type) in each PSEV were examined (Supplementary Fig. 1A–C). The majority of top ranked nodes in a given PSEV were also first neighbor relationships in SPOKE. For example, the multiple sclerosis (MS) *Disease* node is connected to 39 *Anatomy* nodes (such as *MS*-LOCALIZES_DlA-*Central Nervous System*) in SPOKE. Notably, there is 100% overlap between the top 39 ranked *Anatomy* elements in the MS PSEV and all actual MS *Anatomy* neighbors ($n = 39$). Similarly, for *Symptom* nodes connected to MS (such as *MS*-PRESENTS_DpS-*Fatigue*), 80% of first neighbor relationships are maintained in the top n-*Symptom* elements of the MS PSEV. This means that although most of the top nodes were the same, new relationships were prioritized based on the symptoms experienced by individual MS patients at UCSF. Next, the prioritizations of nodes that were not directly connected in SPOKE were considered (Supplementary Fig. 3C). For instance, multiple nodes related to the *response to interleukin-7* were ranked among the top ten *BiologicalProcess* nodes and the node for the *structural constituent of myelin sheath* in the top ten *MolecularFunction* nodes. Though there was an abundance of evidence supporting these relationships, there was neither direct relationship in SPOKE nor was this information stored in the EHRs, thus they must be learned during PSEV creation. These results illustrate the ability of PSEVs to preserve the original information from SPOKE while expanding its significance in a biologically meaningful manner by reaching out to more distant but biologically related nodes. Furthermore, this demonstrates that PSEVs describe each EHR concept in multiple dimensions and is true to the hierarchical organization of complex organisms.

After identifying and implementing a method to embed EHR onto the knowledge network, we sought to verify in a rigorous manner that the obtained vectors were biologically meaningful (i.e., that the expanded set of variables stemming from the EHRs result in a network of related medical concepts). Next, we demonstrate that the PSEV ability to learn genetic relationships can be applied in a high-throughput fashion. In addition, a series

of benchmarks (Supplementary Note 1, Supplementary Figs 4–7) show that PSEVs ability to learn connections can be applied to other edge types, such as *Disease–Disease* (edges from MEDLINE co-occurrence) and *Compound–Compound* similarity (edges DICE similarity), *Compound* to drug–protein (molecular targets; edges from DrugBank, DrugCentral, BindingDB), and *SideEffect-Anatomy* (edges from MEDLINE co-occurrence).

**PSEVs uncover specific *Disease–Gene* relationships**. Because of the multitude of concepts present in SPOKE, multiple paths can connect any two nodes, thus providing redundancy. Thus, we hypothesized that unknown relationships, like the GWAS genes recovered in the high BMI PSEV, could still be inferred even if some of the information was missing because the random walker would traverse similar paths during PSEV computation. To address this point, all of the *Disease–Disease* (e.g., *MS*-RESEMBLES_DrD-*Amyotrophic Lateral Sclerosis*) and *Disease–Gene* edges (*MS*-ASSOCIATES_DaG-*IL7R* and *MS*-DOWN-REGULATES-*PALLD*) in SPOKE were removed and the PSEVs were recomputed the *Disease* PSEVs (PSEV$^{\Delta DD, \Delta DG}$).

The resulting *Disease* PSEVs (PSEV$^{\Delta DD, \Delta DG}$) were visualized in a heatmap and clustered by *Diseases* and *Genes* (Fig. 3a). Clearly defined groups of diseases can be identified in the heatmap, many of which are known to share associated or influential genes. For example, disease Cluster 4 contains mainly neurological diseases, such as multiple sclerosis, Alzheimer's disease, narcolepsy, autistic disorder, and attention-deficit hyperactivity disorder. The Gene cluster most characteristic of Disease Cluster 4 contains 197 genes (Fig. 3b). Within this gene cluster, 96 *Genes* were associated with at least one *Disease* in Disease Cluster 4 (enrichment fold change = 2.0), 33 *Genes* were associated with at least two diseases (enrichment fold change = 3.9), and 15 *Genes* were associated with at least three diseases (enrichment fold change = 5.4; Fig. 3c, d). These results support the hypothesis that PSEVs encode deep biological meaning.

To validate that the recomputed PSEVs (generated without the critical edges) were able to uncover genetic relationships among the complex diseases in SPOKE, a *Disease–Gene* networks (DG) using the top K *Gene* nodes for each *Disease* in PSEV$^{\Delta DD, \Delta DG}$ was created, where K was equal to the number of known gene associations for a given disease. In SPOKE, the ASSOCIATES_-DaG edges represent known associations between *Diseases* and *Genes* and were obtained from the GWAS Catalog[15], DISEASES[16], DisGeNET[17,18], and DOAF[19]. DG networks were generated using either the original PSEVs (DG$^{PSEV}$, blue) or the incomplete, benchmarking PSEV$^{\Delta DD, \Delta DG}$ (DG $^{PSEV\Delta DD, \Delta DG}$, green Fig. 4a). These networks were compared against networks created using three random matrices as a way to generate a null distribution: PSEV$^{RANDOM}$ (DG$^{RANDOM}$, pink distribution Fig. 4a), PSEV-$^{SPOKE SHUFFLED}$ (DG$^{SPOKE SHUFFLE}$, red), and PSEV$^{SEP SHUFFLED}$ (orange, DG$^{SEP SHUFFLE}$). Next, the number of overlapping edges between each of the DG networks and the gold standard *Disease*-ASSOCIATES_DaG-*Gene* (DG$^{SPOKE}$) edges ($n = 12,623$) in SPOKE were compared. When selecting the top K *Genes* using only *Genes* with at least one ASSOCIATES_DaG edge ($n = 5392$), both DG$^{PSEV}$ and DG$^{PSEV\Delta DD, \Delta DG}$ shared significantly more edges with DG$^{SPOKE}$ than with any of the random networks (Fig. 4a; average fold change 15.2 and 2.4 accordingly). This suggests that redundancy in SPOKE paths can be used to infer genetic relationships even when the original (direct) associations are removed.

These results were even more striking when selecting the top K genes using all genes in SPOKE (Fig. 4a inset; $n = 20,945$; average fold change 40.6 and 4.5 accordingly). It should also be noted that, unlike PSEV$^{\Delta DD, \Delta DG}$, both PSEV$^{SEP SHUFFLED}$

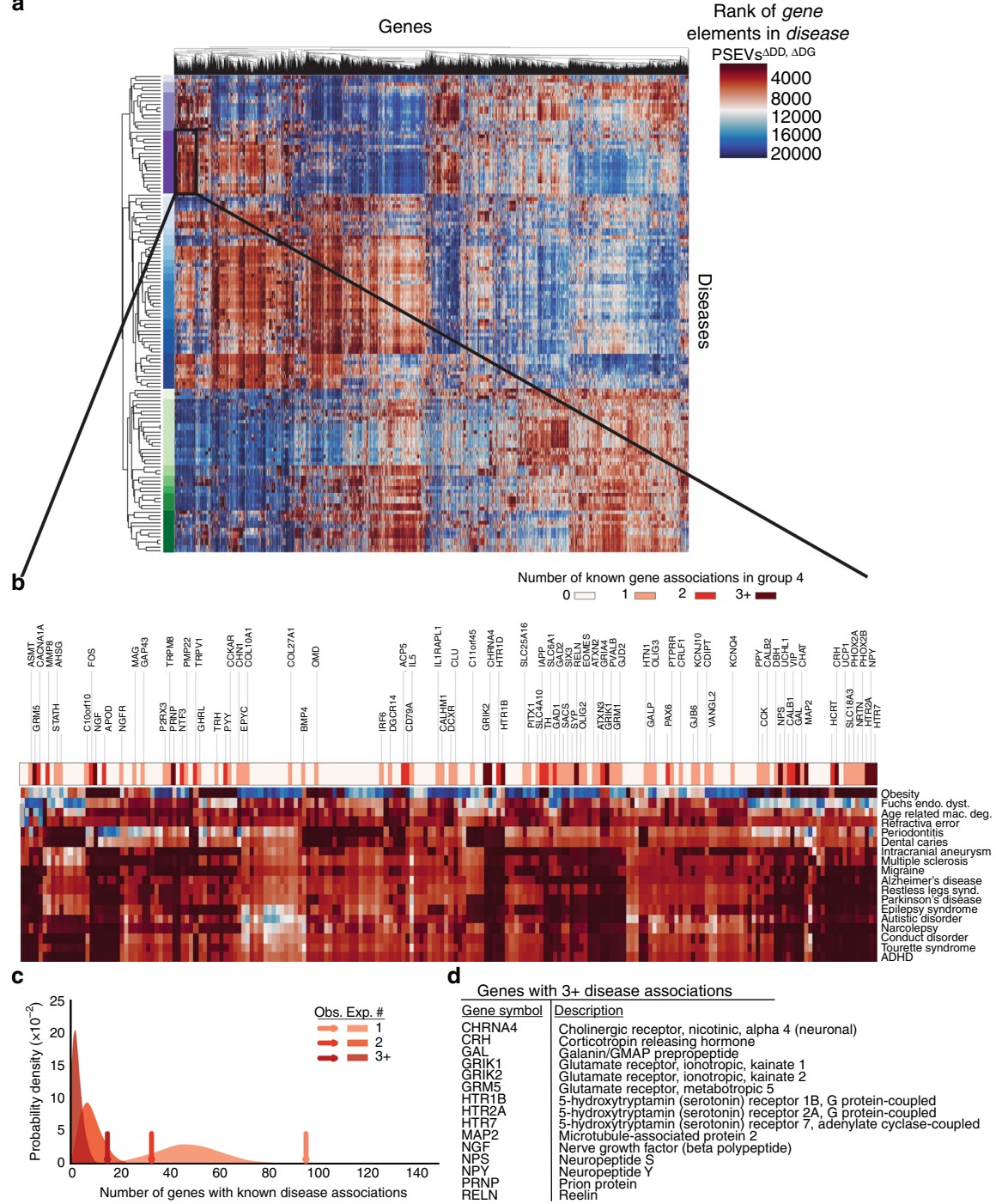

**Fig. 3** Disease cluster by genetic similarity. **a** Heatmap generated with the *Disease* PSEV$^{\Delta DD, \Delta DG}$ (only using elements of *Genes* that associate with at least one disease). The heatmap shows the *Gene* ranks (columns) within each of the 137 *Disease* PSEVs (rows). Both *Diseases* and *Genes* were clustered. *Disease* cluster 4 (*n* = 18) was enriched in neurological diseases and shown in dark purple. **b** Magnification of the 197 *Genes* found in a top *Gene* Cluster (Cluster 6) for *Disease* Cluster 4. Asterisks above gene symbols indicate how many *Diseases* in *Disease* Cluster 4 were associated with that *Gene*. Color bar signifies how many *Diseases* were associated with a given *Gene*. **c** Expected distributions for the number of *Genes* that were associated with at least one, two, or three *Diseases* (1000 random permutations of 18 *Diseases* and 197 *Genes*). Arrows show the observed number over *Genes* within gene Cluster 6 that were associated with at least one, two, or three *Disease* in Disease Cluster 4 and greatly exceed the expected number of *Genes* (fold change = 2.0, 3.9, and 5.4 accordingly). **d** Fifteen*Genes* that were within Gene Cluster 6 were associated with three or more *Diseases* in Disease Cluster 4

and PSEV$^{SPOKE\ SHUFFLED}$ were created without deleting the *Disease–Disease* and *Disease–Gene* edges from SPOKE, therefore the correct edges were present at least some of the time even in the permuted networks, thus providing a higher level of stringency.

**Learning rate differs between edge types.** One of the main challenges with knowledge networks is that as long as our knowledge is incomplete, the networks will suffer from missing edges. The benchmark shown here illustrates the most severe scenario, in which 100% of our knowledge about the relationships

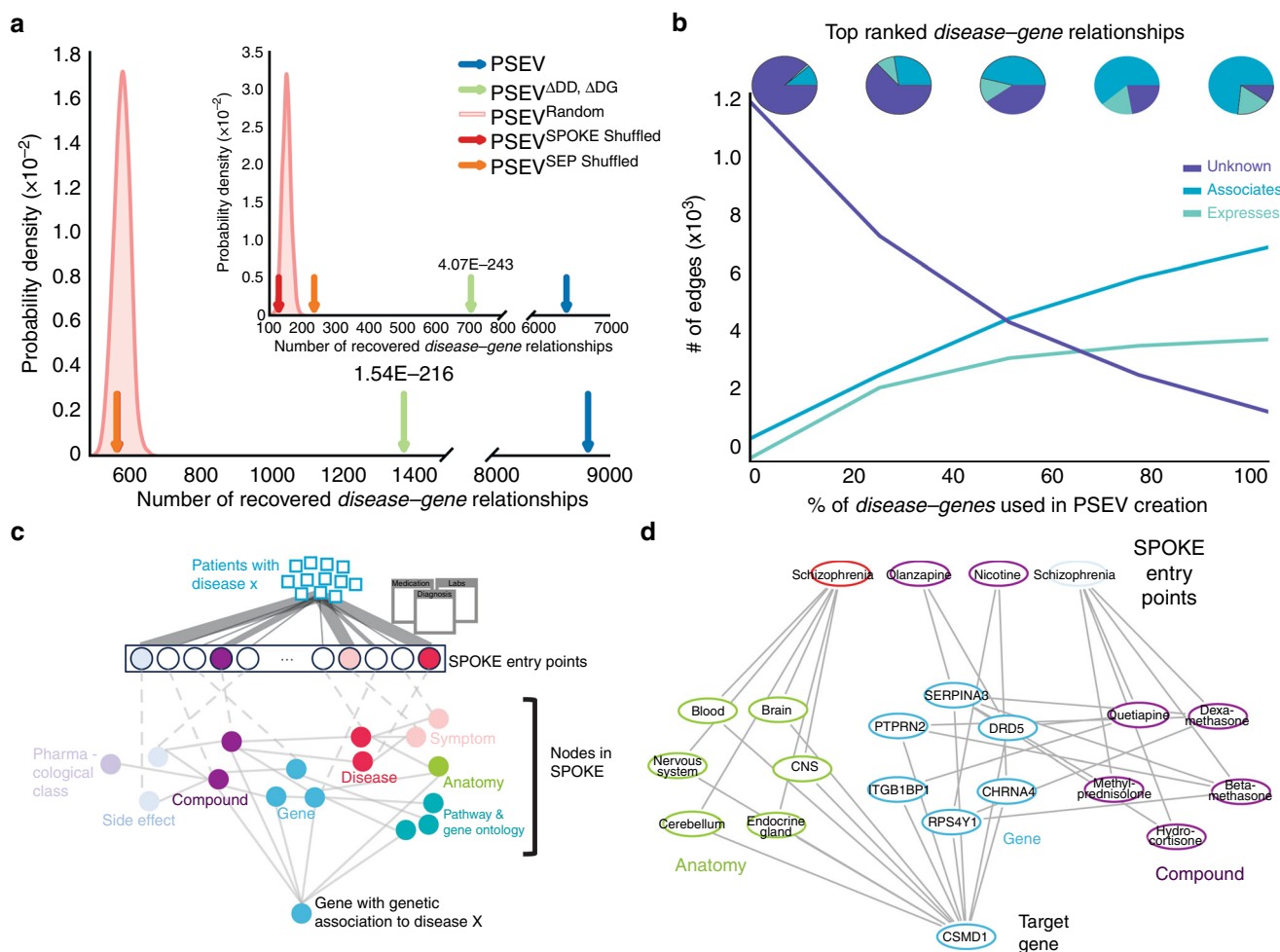

**Fig. 4** Recovering deleted *Disease–Gene* edges. Prior to PSEV$^{\Delta DD, \Delta DG}$ calculation, all of the *Disease–Gene* and *Disease–Disease* edges were deleted from SPOKE. **a** The gold standard *Disease–Gene* network was made from the deleted edges in SPOKE. Plots show the number of *Disease–Gene* relationships using each of the PSEV matrices that overlap with the gold standard networks. The pink distributions show the results from the permuted PSEV matrices (PSEV$^{Random}$; 1000 iterations), while the arrows show the results from the original PSEV (blue), PSEV$^{\Delta DD, \Delta DG}$ (green), PSEV$^{SPOKE\ SHUFFLED}$ (red), and PSEV$^{SEP\ SHUFFLED}$ (orange). **a** The top K genes were selected from the set of genes in the gold standard network or (A inset) the entire set of gene nodes in SPOKE. **b** The breakdown of top disease–gene relationships as knowledge (edges) were added back to the network. **c** To uncover how the deleted *Disease–Gene* associations were recovered using the PSEVs, we retraced the shortest path between the most important SPOKE Entry points (SEPs) and the desired Gene. Patients with *Disease* X put pressure on the SEPs (top). The SEPs that receive the most significant amount of pressure are colored by node type. Information then flows through other nodes in SPOKE (middle) before reaching the *Gene* that was genetically associated to *Disease* X (bottom). **d** In the GWAS catalog, schizophrenia and CSMD1 are associated. As outlined in **b**, the information flows from the significant SEPs of patients with schizophrenia to CSMD1

among *Diseases* and between *Diseases* and *Genes* is removed. To evaluate performance of the algorithm as the network gains knowledge, edges were slowly added back to the network. We found that the PSEVs learned well-established (ASSOCIATES from the GWAS Catalog, DisGeNET, DISEASES, or DOAF) *Disease–Gene* edges before the noisier (UP(DOWN)REGULATES from StarGEO) edges (Fig. 4b). This is most likely due to the fact that well-established (associated) *Genes* are necessarily drivers of (not reacting to) a *Disease*. In practice, this would cause the random walker keep going back to *BiologicalProcess, CellularComponent, MolecularFunction*, and *Pathway* nodes that are important for a given *Disease* and thereby push information to *Genes* involved in those activities. Alternatively, the random walker could travel to *Anatomy* nodes that express *Genes* that are associated with a *Disease* or through *Compounds* that are used to treat (or even those that exacerbate) a *Disease*. This further demonstrates that the relationships inferred within PSEVs are biologically meaningful.

**Retracing the path between SEP and genes**. Finally, to understand how the patient population at UCSF influenced the PSEVs to correctly rank *Disease–Gene* associations, the shortest paths were retraced between the significant SEPs of a given *Disease* and the associated *Gene* (Fig. 4c; Methods). For example, the locus containing *CSMD1* is associated with schizophrenia in the GWAS Catalog. Figure 4d shows why the gene *CSMD1* was one of the top ranked genes in the PSEV$^{\Delta DD, \Delta DG}$ for schizophrenia. The weight from the EHRs of schizophrenia patients at UCSF drives information toward *Anatomy*, in which *CSMD1* is expressed or regulated and *Compounds* that bind or regulate *Genes* that interact or regulate with *CSMD1*. The combined weight highlights *CSMD1* as a gene that is associated with schizophrenia. This example highlights the fact that inferences made with this method are not black box predictions, but the information used to make the inference can be traced back to the exact concepts. We believe that knowledge-based clear-box algorithms, such as the one presented here, will be pivotal in the advancement of precision medicine.

## Discussion

Uncovering how different biomedical entities are related to each other is essential for speeding up the transformation between basic research and patient care. When deciding the best therapeutic management strategy for a patient, physicians often need to think about the symptoms they present, their internal biochemistry, and potential molecular impact and adverse events of drugs simultaneously. A well-trained and experienced doctor will likely prescribe the best course of action for that patient. However, significant heterogeneity is seen even across the best hospitals on what best course of action means for a given patient, resulting in poor consistency, a labyrinth of solutions, and ultimately lack of evidence-based medicine. Since it is naturally impossible for a single person to retain and recall all the necessary and relevant information, an efficient manner to incorporate this knowledge into the health-care system is needed. We argue that since PSEVs can be created for any code or concept in the EHRs, it is possible they could provide such solution. Using PSEVs, we were able to integrate what we have learned from the last five decades of biomedical research into the codes used to describe patients in the EHRs. As a result, these embeddings serve as a first step to bridging the divide between basic science and patient data.

Our method for the integration of EHRs and a comprehensive biomedical knowledge network is based on random walk. Random walk has been applied to a wide variety of biological topics, such as protein–protein interaction networks[20], gene enrichment analysis[21], and ranking disease genes[22–24]. In addition, random walk has been used to infer missing relationships in large incomplete knowledge bases[25]. Our method includes the generation of PSEVs, as a way to embed medical concepts onto the network. The entire patient population at UCSF was used to determine how important each node in SPOKE is for a particular code. Therefore, each PSEV describes EHR codes in both a high-level phenotypic and deeper biological manner.

We demonstrated that not only do PSEVs carry the original relationships in SPOKE, but also were able to infer new connections. This was illustrated by ability of PSEVs to recover deleted *Disease–Disease*, *Disease–Gene*, *Compound–Compound*, and *Compound–Gene* edges as well as to infer new relationships between *SideEffect* and *Anatomy* nodes. Other than just showing that PSEVs can learn relationships between different types of nodes, these tests illustrated that PSEVs can learn relationships between nodes at a variety of lengths apart from one another. By correctly inferring the *Disease–Gene* and *Compound–Gene* edges, we demonstrated that PSEVs could uncover higher-order relationships, such as those between a cohort and SPOKE. At the same time, correctly inferring *Disease–Disease* and *Compound–Compound* edges demonstrated that PSEVs could uncover relationships among EHR concepts themselves. Finally, by inferring *SideEffect-Anatomy* edges, we proved PSEVs could find SPOKE-level relationships. These tests served as our proof of principle that PSEVs can learn multiple types of new relationships.

Furthermore, these results illustrate that, unlike black box methods, PSEVs are capable of embedding phenotypic traits, such as risks, co-morbidities, and symptoms. Other vectorization methods like word2vec are able to learn relationships, however, since the elements within the vector are unknown they cannot be traced back to a given trait in the EHRs. Similarly, though it is possible to identify these phenotypic traits using a statistical analysis of a single cohort, the benefit to using PSEVs is that these traits are identified in a high-throughput fashion for every concept in the EHRs and outputs them in a format that can be used in machine-learning platforms. PSEVs, and other clear-box algorithms, allow us to integrate knowledge into data, therefore generating deeper, informed characterizations that can be understood by both humans and machines.

The main limitations of this approach mostly stem from the potential inaccuracies in the EHRs and the incompleteness of the knowledge networks (SPOKE). First, while maintaining the trust and privacy of patients remains paramount, it has also made it difficult for institutions to share even de-identified records. Not being able to openly share data means that the patient population used may not be representative of the general population, especially in terms of race, ethnicity, education, and income. Second, many institutions do not use standardized medical terminology, thus making it challenging to accurately map EHR concepts to SPOKE. That being said, institutions that use EHR formats that utilize standard terminologies, like the Observational Medical Outcomes Partnership (OMOP) Common Data Model, can easily implement this in their own system. While we did not use OMOP in this work, efforts by our group and others are ongoing in this direction. Finally, we are limited by the fact that as long as our biomedical knowledge is incomplete, the same will be true for our knowledge networks. In this regard, SPOKE is continually under development and future versions will increase in complexity and completeness. However, our results show that adding context with the EHRs actually enabled us to learn new relationship in the network, thereby growing our biomedical knowledge. We believe that these limitations are inherent to this field of study and that the development of tools, such as the one presented here, can spur collaboration between institutions and help overcome these limitations.

The potential uses of PSEVs are vast. We recognize that several associations in EHRs can be uncovered using clinical features alone, and several machine-learning approaches are already being utilized to that end[26]. However, since PSEVs describe clinical features on a deeper biological level, they can be used to explain why the association is occurring in terms of Genes, Pathways, or any other nodes in a large knowledge network, like SPOKE. Consequently, PSEVs can be paired with machine learning to discover new disease biomarkers, characterize patients, and drug repurposing. With implementation of some of these features, we anticipate that PSEVs or similar methods will constitute a critical tool in advancing precision medicine.

## Methods

**Electronic health records**. UCSF supplied the EHRs in this paper through the Bakar Computational Health Sciences Institute. Almost one million people visited UCSF between 2011 and 2017. Out of 878,479 patients, 292,753 had at least one of the 137 complex diseases currently represented in SPOKE. The EHRs were de-identified to protect patients' privacy. No IRB approval was required for this research. For this paper, we collected the information on the cohort of patients with complex diseases using de-identified LAB, MEDICATION_ORDERS, and DIAGNOSES tables. The LAB table contains the lab test orders and results, including the actual measurements and the judgment of whether the results were abnormal. The MEDICATION_ORDERS table contains prescriptions with dose, duration, and unit. The DIAGNOSES table contains diagnosis and symptoms with ICD9 and ICD10 codes. These tables are linked by Patient IDs (one unique ID for each patient) and Encounter IDs (one unique ID for each encounter a given patient has with our medical system).

**Scalable precision medicine oriented knowledge engine**. SPOKE is a heterogeneous knowledge network that includes data from 29 publicly available databases, representing a significant proportion of information gathered over five decades of biomedical research[12]. This paper was powered by the first version of SPOKE, which contains over 47,000 nodes of 11 types and 2.25 million edges of 24 types. The nodes (Anatomy, BiologicalProcess, CellularComponent, Compound, Disease, Gene, PharamacologicalClass, SideEffect, and Symptom) all use standardized terminologies and were derived from five different ontologies. The sources and counts of each node and edge type are detailed in Supplementary Tables 1 and 2.

**Connecting EHRs To SPOKE**. EHRs were connected to SPOKE *Disease*, *Symptom*, *SideEffect*, *Compound*, and *Gene* nodes. To connect to *Disease* nodes, ICD9/10[27] codes in the EHRs were translated to *Disease* Ontology identifiers[28,29]. Since this relationship was used to select the patient cohort, we manually curated the mappings. The connection to *Symptom* and *SideEffect* nodes was also made from translating the ICD9/10 codes via MeSH identifiers and CUI, respectively. The

relationship between *Compound* nodes and EHRs was derived by mapping RxNorm to the FDA-SRS UNIIs (unique ingredient identifiers) to DrugBank Identifiers. Lab tests were connected to multiple node types in SPOKE using the Unified Medical Language System (UMLS) Metathesaurus[30]. The LOINC[31] codes in the EHRs were mapped to CUI and then mapped to a second CUI (CUI2) using UMLS relationships. A connection between LOINC and SPOKE would be made if CUI2 could be translated to a node in SPOKE. CUIs with nonspecific relationships were excluded. From the UCSF EHRs, 10,499 unique codes, found in the Diagnosis, Medication Orders, and Labs tables, were mapped to 3527 nodes in SPOKE. Of these, 3233 were seen in the complex disease cohort and were used as the SEPs.

**Generating PSEVs**. First, we initialized a $n \times n$ SEP transition matrix (where $n =$ the number of SEPs) and set every value to zero. Then for each patient in the complex disease cohort, we created a binary vector of the SEPs in their EHRs and divided it by the sum of the vector. This patient vector was then added to the rows of the SEP transition matrix that corresponded to the SEPs found in the patient's EHRs. Once every patient was accounted for, the SEP transition matrix was transposed and divided by the sum of the columns.

Next, we made an adjacency matrix using the edges in SPOKE to create a SPOKE transition probability matrix (TPM), in which each column sums to 1. The SPOKE TPM was then multiplied by $1-\beta$ where $\beta$ equals the probability of random jump. An extra row was then added to the SPOKE TPM and filled with $\beta$.

Last, the PSEVs were generated using a modified version of the PageRank algorithm[13,14]. In this version of PageRank, for each PSEV, the random walker traverses the edges of SPOKE until randomly jumping out of SPOKE (at probability $\beta$) to the given SEP. The walker will then enter back into SPOKE through any SEP using the probabilities found in the corresponding column of the SEP transition matrix. The walker will continue this cycle until the difference between the rank vector in the current cycle and the previous cycle is less than or equal to a threshold ($\alpha$). The final rank vector is the PSEV and contains a value for every node in SPOKE that is equivalent to the amount of time the walker spent on each given node.

Genes were selected from the GWAS Catalog if they were associated with an increase in BMI and were genome wide significant.

**Generating *Disease* PSEV matrix for benchmark**. We created *Disease* benchmark PSEV matrix ($\text{PSEV}^{\Delta DD, \, \Delta DG}$) by removing the *Disease–Disease* and *Disease–Gene* relationships in SPOKE prior to PSEV creation. We then used z-scores to normalize the $\text{PSEV}^{\Delta DD, \, \Delta DG}$ and ranked the elements for each type of node.

**Random *Disease* matrix**. In order to test the importance of the edges between SEPs and SPOKE as well as SPOKE's internal edges, we generated three types of random PSEVs. First, we created a completely random PSEV matrix by using the Fisher–Yates method to permute the SPOKE nodes for each *Disease* PSEV ($\text{PSEV}^{\text{random}}$). Second, for each edge type in SPOKE, we randomly shuffled the edges prior to PSEV creation ($\text{PSEV}^{\text{shuffled\_SPOKE}}$). Third, we shuffled the edges between the SEPs and SPOKE prior to PSEV creation ($\text{PSEV}^{\text{shuffled\_SEP}}$). It should be noted that when creating $\text{PSEV}^{\text{shuffled\_SEP}}$, all SPOKE relationships were maintained. In addition, SEP-SPOKE edges were only shuffled once and therefore any relationships coming directly from the merged EHRs to the SEPs would be conserved. Once random PSEVs were created, they were normalized using z-scores.

**Inferring *Disease–Gene* relationships from PSEVs**. In addition to looking at *Disease–Disease* relationships, we examined the ability of PSEVs to rank the *Disease*-ASSOCIATES_DaG-*Gene* relationships from SPOKE. The Disease-ASSO-CIATES_DaG-Gene edges ($n = 12,623$) in SPOKE come from four sources: the GWAS Catalog[15], DISEASES[16], DisGeNET[17,18], and DOAF[19].

After z-score normalizing the PSEV matrix, within each Disease *PSEV, Gene*s were ranked 1 to 5392 or 20,945 when using only *Gene*s that are associated with at least one *Disease* or the full set of *Gene*s accordingly, such that a *Gene* ranked 1 would denote the most important *Gene* for a given *Disease* based on the PSEV matrix. Then for each *Disease* PSEV, K Genes were selected where K was equal to the number of *Genes* are associated with a given *Disease*. The *p*-values (binomial test) for ability of each *Disease* PSEV to correctly rank, the associated *Genes* were then combined using Fisher's method[32]. This evaluation was applied to the original PSEV, benchmark PSEV, and all three random networks (Fig. 4a, b).

**Creating Disease–Gene heatmap**. The $\text{PSEV}^{\Delta DD, \, \Delta DG}$ matrix was filtered such that it only contained Disease PSEVs and the gene elements that are associated with at least one disease in SPOKE ($m = 137$, $n = 5392$). This was then used as input into the seaborn clustermap package in python with the settings method = "average" and metric = "euclidean". Here, method refers to the method used to calculate the linkage and metric is the way in which we calculate the distance within the method.

**Shortest paths between SEP to target nodes**. To understand how the PSEVs were able to recover deleted relationships, we traced from the target node back to the contributions of each SEP. To achieve this, we z-score normalized the original

SEP transition matrix used to calculate the PSEVs. Then we created a SPOKE only PSEV matrix ($\text{PSEV}^{\text{SPOKE-only}}$) that forces the random walker to randomly restart ($B = 0.33$) from a single SEP. The $\text{PSEV}^{\text{SPOKE-only}}$ matrix was created using SPOKE with deleted *Disease–Disease* and *Disease–Gene* edges or *Compound–Compound* and *Compound–Gene* edges when recovering the paths for $\text{PSEV}^{\Delta DD, \, \Delta DG}$ and $\text{PSEV}^{\Delta CC, \, \Delta CG}$ accordingly. The $\text{PSEV}^{\text{SPOKE-only}}$ matrix allows to identify the contribution of an individual SEP to any of the downstream nodes. We then took the product of a given *Disease* or *Compound* transposed vector from the SEP transition matrix with the $\text{PSEV}^{\text{SPOKE-only}}$ to generate contributions of each SEP to the target node. The most important SEP were selected if they were in the top 0.1 percentile of contributors. We then found the shortest paths between the important SEPs and the target node.

**Reporting summary**. Further information on research design is available in the Nature Research Reporting Summary linked to this article.

## Data availability
The version of SPOKE used in the research is available on neo4j and github. The PSEVs are available on the PSEVexplorer and github.

## Code availability
The code used to create these PSEVs, as well as some example patient data, is available on github. The code is written in python. Please read the README for information on downloading and running the code.

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

## Acknowledgements

Use of UCSF de-identified data was made possible by Dana Ludwig MD and the UCSF Information Technology Services Academic Research Systems team and the Clinical Data Research Consultations service. We thank Sourav Bandyopadhyay, Riley Bove, Jeffrey Gelfand, Sharat Israni, and Keith Yamamoto for helpful discussions. Partial support for this work was provided by grants from Genentech to A.J.B. and S.E.B (G-54860). The sponsor had no role in the design or implementation of this study. Research reported in this publication was supported by funding from the UCSF Bakar Computational Health Sciences Institute and the National Center for Advancing Translational Sciences of the National Institutes of Health under award number UL1 TR001872. In addition, we would like to thank Achievement Rewards for College Scientists (ARCS) Scholarship and the NHI BMI Training Grant (T32 GM067547/ 4T32GM067547–14). S.E.B. holds the Heidrich Family and Friends Endowed Chair and the Distinguished Professorship in Neurology I at UCSF.

## Author contributions

C.N. and S.E.B. conceived the idea. C.N. wrote the code and performed all analyses. S.E. B. provided guidance and supervised the research. A.B. provided guidance on the project. C.N, A.B., and S.E.B. wrote the paper.

## Additional information

**Competing interests:** C.N. and S.E.B declare no competing interests. A.B. reports grants and nonfinancial support from Progenity, personal fees from NuMedii, personal fees from Personalis, grants and personal fees from NIH (multiple institutes), grants from L'Oreal, grants and personal fees from Genentech, personal fees from Merck, personal fees from Lilly, personal fees from Assay Depot, personal fees from Geisinger Health, personal fees from GNS Healthcare, personal fees from uBiome, personal fees from Roche, personal fees from Wilson Sonsini Goodrich & Rosati, personal fees from Orrick, Herrington & Sutcliffe, personal fees from Verinata, personal fees from 10x Genomics, personal fees from Pathway Genomics, personal fees from Guardant Health, personal fees from Gerson Lehrman Group, personal fees from Nuna Health, personal fees from Samsung, personal fees from Milken Institute, personal fees from Brown University, personal fees from Oregon Health Sciences University Knight Cancer Center, personal fees from Vermont Oxford Network, personal fees from University of Chicago, personal fees from Mount Sinai School of Medicine, personal fees from University of Pittsburgh School of Medicine, personal fees from Capital Royalty Group, personal fees from Champalimaud Foundation, personal fees from Scripps Translational Science Institute, personal fees from Washington University in St. Louis, personal fees from University of Maryland, personal fees from HIMSS, personal fees from Federation of Clinical Immunology Societies (FOCIS), personal fees from Kansas City Area Life Sciences Institute, personal fees from Association for Molecular Pathology, personal fees from TEDMED, personal fees from Physician's Education Resource, personal fees from Optum Labs, personal fees from National Jewish Health, personal fees from Federation of the Israeli Societies for Experimental Biology (FISEB), personal fees from Pfizer, personal fees from Bayer, personal fees from Fusion Conferences, personal fees from Accelerating Biopharmaceutical Development (AccBio), personal fees from Three Lakes Partners, personal fees from Pediatric Academic Societies, personal fees from Korean Society for Biochemistry and Molecular Biology, personal fees from Human Proteomics Organization (HUPO), personal fees from HudsonAlpha, personal fees from Tensegrity, grants from Intervalien Foundation, personal fees from Association for Academic Health Sciences Libraries, personal fees from Westat, personal fees from FH Foundation, personal fees from University of Kentucky, personal fees from University of Pennsylvania, personal fees from The Transplantation Society, personal fees from California Office of Planning and Research, personal fees from WuXi, personal fees from University of Arkansas, personal fees from FlareCapital, personal fees from National Academies, personal fees from Helix, personal fees from American Urological Association, personal fees from Association for American Medical Colleges, personal fees from Roam Insights, personal fees from United Network for Organ Sharing, personal fees from American Association of Allergy Asthma and Immunology, personal fees from University of Michigan, personal fees from Autodesk, personal fees from Regenstrief Institute, personal fees from American Medical Association, personal fees from Precision Medicine World Conference, personal fees from University of Chicago, personal fees from Mars, and personal fees from Kneed Media. Stanford University pays royalties each year on licensed intellectual property. MIT Press pays royalties each year on book sales.

