## [Peer Review File · Nature Communications]

Reviewers' Comments:

Reviewer #1:

Remarks to the Author:

This ms. by Nelson et al. builds on the previous work of the Lab of Sergio Baranzini that established a big heterogeneous knowledge-graph (network) in which medical concepts (GENES, DISEASES, DRUGS...) represent the nodes, with their relationships (DRUG A used in DISEASE B) known from existing databases, represented by the edges. The central advance now is that the importance of these nodes of SPOKE, and consequently, their relationships is informed by “real world” data from electronic health records (EHR). A concept X in the EHR data (e.g., a disease X) that defines a patient cohort, is associated with a vector called PSEV (Propagated SPOKE Entry Vectors) whose vector elements assign to the SPOKE nodes a relative weight (or RANK) indicating how these node concepts are associated with the disease X by incorporating information on the co-occurrence of SPOKE node concepts (e.g. symptoms, drugs, genetic variants) with those present in a given cohort of patients with disease X that is documented in the EHR. The further association of node concepts in SPOKE that are NOT contained in the EHR (but part of the PSEV) is achieved by an algorithm of a random walk on the SPOKE network that is constrained by the EHR cohort information, a process inspired by the PageRank algorithm that assigns importance to websites based on surfer behavior (hence the term Propagated in PSEV). The authors then demonstrate the power of this integration between the SPOKE knowledge graph and patient EHR data using the examples of obesity, multiple sclerosis and schizophrenia and found that this new integration of real world data of EHR with existing knowledge on relationships between medical concepts from databases summarized in the SPOKE knowledge graph can produce NEW relationships between medical concepts.

This work is a major tour-de-force effort that is most impressive and perhaps the most advanced that I have seen at this highly active front of advance into digital medicine. The overarching concept is straightforward, intuitive yet very original. It constitutes an elegant way to merge established database knowledge with “fresh” patient information from large patient cohorts.

I am genuinely excited about this piece of work and expect that the visionary members of the community dealing with digital health and also acting as practioners will readily see the potential value of this work.

Of course such an ambitious undertaking cannot avoid the exposure of limitations in its product and there is indeed room for improvement – but that is part of the natural course of true innovative progress. The functional shortcomings that the inquisitive reader will certainly spot shall not prevent publication of this work, for they are those kind of limitations that will spark new ideas for improvement – ideas that would have never surfaced without the catalyzing effect of this elegant and rich piece of work. Also, one can readily fathom additional analysis. One may for instance ask how the performance of the SPOKE/PSEV approach will look like beyond the examples used, such as obesity or MS which are well defined, robust concepts. In fact the global robustness has yet to be demonstrated – but this task, I would contend, requires large-scale approaches and should not be part of the authors’ work but rather left to the community of early adopter users who hopefully will further scrutinize the platform and offer constructive criticism and suggestions in the near future.

In fact, the imaginative and innovative reader will by reading of this work, despite the limitations (or precisely BECAUSE of them) be already enticed to come up with an array of new ideas how to take the platform to the next stage. Therefore, I think this ms. is sufficiently mature and solid and certainly of great interest so as to warrant publication without major additional corrective analyses.

However, for this ms to serve its purpose and have the impact it deserves, the presentation and notably the didactical explanation of a series of abstract concepts that the author rely upon (and too much take for granted) must be improved. Below I list in detail the opportunities to improve clarity and make this ms. more easy to comprehend and hope that the authors agree with most of them.

=====

[1] Ln 94, Ln 103: MORE CLARITY ON PRESENTING KEY CONCEPTS WOULD BE GREAT! For readers without previous exposure to SPOKE I am afraid the result section's introduction of the central terms 'PSEV', 'SEP', 'HER concept' etc is rather a steep learning curve. It would facilitate understanding if the authors more explicitly describe the elementary (ontological) essence of these key elements, e.g. What kind of object (a vector!) is PSEV and of what (a concept??) is it a property? I know all the information is somehow implicitly there (e.g. V stands for vector) but rarely explicitly exposed and explained.

It sounds as if 'PSEV' represents (is a property of) a concept in the EHR (as repeated again in Ln 152), where a concept (in feature space) in the EHR can be a variable that is orthogonal (can be assigned) to any patient, such as BMI, a gene locus, a disease, a drug being taken by said patients.... (please name examples upfront already in Ln 94 when explaining). Having this concrete examples would help greatly! Also please explicitly say that PSEV has the form of a vector that has as its N elements the weights assigned to each node in SPOKE. Also say what is N (the number of nodes in SPOKE?)

From the above explanation in the opening of the RESULT section, it also sounds as if the category to which the NODES in SPOKE belong and the category to which "CONCEPTS" in EHR (medical terms, that can be assigned to patients...] are one and the same and hence are allowed overlap? Is it fair to say that hence, the term 'CONCEPT' can refer to both, nodes in SPOKE and CONCEPTS *sensu strictiore* (medical variables...) in the EHR? This is the impression one gets.

[2] But then there is a subtlety that becomes apparent from the first example and that questions the validity of the definition of PSEV in [1]: It turns out that the notion of mapping of EHR CONCEPTs to a PSEV is actually not precisely as defined in the text (Ln 94). The mapping is not between an EHR CONCEPT to PSEV, but between a VALUE (RANGE) *within* an EHR CONCEPT – one that defines a COHORT (a "natural" subpopulation). Thus, ro be specific, 'BMI' is not the concept that maps into a PSEV but rather, a given *value* of BMI, i.e., what maps to a PSEV is not a CONCEPT (variable) but a particular instance (value) of a variable, such as a range of the BMI, e.g. Obesity **Cohort IV**.

*This is CRUCIAL because this is where the biological meaning of a PSEV comes from: A PSEV is **COHORT-SPECIFIC**, where a cohort is defined by a specific natural property (patients having a particular disease X) WITH respect to a CONCEPT (disease X).*

I think this relationship (mapping) should be emphasized and also indicated in the FIG 1, e.g. with a double-headed long arrow connecting the text "BMI Cohort IV" on the top with the text "Propagated SPOKE Entry vector" in the bottom. This would make a lot of things clearer.

[3] Ln 105: The definition of SEPs is also semasiologically not kosher: "Each structured EHR table contains codes that can be linked to standardized medical terminology allowing direct links to SPOKE, referred to as SPOKE Entry Points (SEPs)." → SEPs are NOT '*links*' as this sentence implies; but an SEP,

as I understand it, is a 'node' in SPOKE; more specifically it is a SUBSET of the set of all the nodes in SPOKE, defined by mapping to a concept that exists in the EHR. Is this correct?

Similarly fuzzy expression about the essence of SEPs is the statement Ln369: "PSEVs could find SEP to SPOKE level relationships". Please avoid what philosophers call "category mistakes".

[4] The FIGURE 1B offers a much needed help for understanding the concepts but it could be improved in some points:

-4a- Please indicate (label) with an arrow pointing to a specific element in the graph, what is the actual entry points (SEP), namely: the colored circles

-4b- Also use a label/arrow to indicate that the entire horizontal bar at the bottom (step 4) is the PSEV.

-4c- With another label/arrow describe what a small square in that PSEV bar represents: Do they represent ALL the nodes of SPOKE (each square = one node), with the white ones representing nodes that are NOT SEPs and hence have no weight (weight = 0)? (Just my educated guess). It would help to repeat that the PSEV is essentially a weighted vector that lists all the nodes in SPOKE (if that is the case).

-4d- Another label/arrow is needed that explains that each of the green empty square in Step 1 represents a patient (I believe) of that one given cohort.

-4e- It may help to show in the box "UCSF EHRs" a simple cartoon of an entire EHR table with patients as rows and CONCEPTS has column heading, of which one is BMI, and label the columns as "CONCEPT and connect one of them to one of the colored circle in the SEP vector of Step 1.

-4f- The label 'SEPS' should be 'SEPs' (plural of SEP)

[5] Ln 126: Not sure why k-means is used to generate the four classes of BMI. It makes sense to not use "a priori" defined classes which I guess are defined by numerical ranges of BMI and not shown in the graph. By contrast the authors observed four "peaks" (subpopulations) in their 800K UCSF patients, shown in Fig 1A. Are these the same as the "a priori defined"? Why not used four natural subpopulations? The fact that the k-means group boundaries do not coincide with the four clearly distinguishable natural subpopulations in the multi-modal distribution of the UCSF cohort means that the standard deviations of the individual subpopulations is not invariant. I am not sure if by "a priori" the authors mean (i) these four peaks from the data or (ii) some numerical ranges defined elsewhere. In any case, the current description in the text is not logically coherent but could readily be corrected by saying what is meant by "a priori".

If correcting this point, please also modify the text at Ln186 to make it consistent.

[6] Ln131: The PageRank algorithms is actually not well explained. The text points to FIG 1B which does not provide a good explanation, and the legend is very cursory in this respect. What does (Ln 136) "jump back to the patient population with probability beta" mean? I think one can only jump back to a node, not to a 'population'. (This is one of many mild category mistakes in this ms). Do you mean jump back to a node that is an SEP which is linked to a concept in the specific COHORT (hence a 'population')? In the Methods on p.17, Ln459 there is a differing –perhaps better– explanation for what is the probability of beta: "...for jumping out of SPOKE to the given SEP". This sounds more precise but what is a "given SEP"? Is it what I suspect in the previous sentence? What does "out of SPOKE" mean? –Do you mean the SPOKE nodes that do NOT represent SEPs??

But in Ln459 in the METHODS, beta is defined as the "probability for a random jump". (ANY jump?) So what is beta?

It is also difficult to relate beta to anything in the original PageRank and the reference to the original papers are not very helpful. Is the random walk the equivalent to the "random surfer" and is (beta-1) sort of like the damping factor d in the original PageRank algorithm?

[7] Another related unclear explanation is the term ‘rank vector’ that results from the random walk (Ln137) . Is this essentially the PSEV? Or how do you get to the PSEV from here?

Until now, the reader is left with the impression (through guessing) that the PSEV is a vector that contains N elements (for the N nodes in the SPOKE) with the values representing some “weight” indicating the importance of a node-concept in SPOKE for THE EHR-concept (disease X of a cohort) to which the PSEV is associated to. This is important for later when “ranks” of diseases are compared for the various PSEVs of the BMI categories (FIGURE 2A). Please articulate all these relationships more explicitly and clearly. This relates to my points [1] and [2].

[8] Ln 152: It will help for understanding if the authors take an opportunity to restate the definition of PSEV by modifying the sentence here to (if I am correct): “The final PSEV is representative of how important EACH node of SPOKE is, not just the subset of nodes that comprise the SEPs.”

This would also confirm that the number N of components of the vector PSEV vector is the same as the number of nodes N in SPOKE. The reader had to assume this all along since it is not explicitly articulated (although inferable).

It may also help here to restate that each BMI category (I, II, III, IV) has its own PSEV – this would help the comprehension of FIG 2A (since so far PSEV has been, perhaps sloppily, equated to a EHR concept, and not a subclass within a concept (defined by a cohort), as discussed above, point [2])

[9] Ln163 on: What is “average BMI?” The FIGURE 2A (which is hard to read) relates to the 4 classes of BMI. Why not say here: “To learn more about the relationships between the four categories of BMI (I=Underweight, II=normal weight, III=Overweight, IV=Obese) and these potentially associated diseases, we plotted for each of these categories the rank of these Disease elements in the four respective PSEVs (Figure 2A)”.

Then please improve FIGURE 2A:

-9a- Add the labels I, II, III, IV to the X-axis categories. It is confusing that the authors suddenly switch to the names of these categories without mentioning the equivalence

-9b- The legend of INSET (I guess the diseases chosen) is too small – UNREADABLE!

-9c- The Y-axis should say “Rank of Disease in the PSEVs”

[10] Ln193 etc: The sloppily metaphoric verbiage, such as “hypertension MOVES...” or a disease undergo “Rank change” is confusing. I assume all this is related to moving from lower to higher BMI categories. Please say so explicitly – the reader has not internalized the framework that the authors have in mind when they use the terminology of “movements” in this purely static analysis

[11] Ln192 ff: For the discussion of association with GENOTYPE – the terminology has to be more precise. What does it mean “Genes that are positively correlated with average BMI”. I assume every gene discussed is present in everybody. Also what does “average” BMI means? Within a category? What is a GENE?? Shouldn’t it say, instead: “gene LOCI for which there is a variant (SNP) that is associated with BMI...”?

[12] Ln205: Moreover, this same section, entitled “...Learn Genotypic Traits..”, suddenly sways off the topic and discusses ALTERED GENE EXPRESSION. The concept of gene expression is part of the phenotype and is not necessarily directly linked to genotype.

[13] Ln 221: “If the top 39...” should start a new sentence.

[14] Ln 255: In moving towards the heatmap, the authors utilize the concept of the modality of the edges in SPOKE, but have not explicitly introduced it in this ms. other than mentioning the number of 24 TYPES of edges in the INTRODUCTION. As said, I would not rely on readers being already familiar with SPOKE. Somewhere, perhaps here, the nomenclature and essence of edges should be briefly defined. How is a Disease-Disease or a Disease-Gene link established in SPOKE? (Give example) What does “ASSOCIATES-DaG” (Ln273) and what does “REGULATES” (Ln303) mean exactly?

[15] FIGURE 3A: More details are needed here too. How the heatmap was constructed should be explained in more detail. It is essentially a bi-clustering of a data in an $M \times N$ table, eg with rows = diseases and columns = genes. But how were they selected, or extracted, from the PSEVs?? The Legends is better than the text but still not sufficient. Also please label the two axes (ROWS vs COLUMNS) of the heat map with: ‘GENES’, ‘DISEASES’ (this always helps even if it is in the legend)

[16] Ln303: Why is the edge “REGULATES” (whatever it means, see comment above) more “noisy” than “ASSOCIATES” given that the former is a molecular interaction and the latter just a cohort observation?

[17] DISCUSSION. While I am empathetic to the author’s enthusiasms about the potential utility of their platform, and prefer short discussions (as it is now), I think that one paragraph on (current) limitations leading to an outlook (where precisely one could improve) would be in order.

Reviewer #2:

Remarks to the Author:

Summary:

The authors present an interesting and very comprehensive analysis which combines a knowledge base of literature curated biomedical data with electronic health records. The ultimate goal of the study is to aid in the implementation of precision medicine by viewing individual patients in the context of the vast amount of existing biomedical data and knowledge. Connecting basic biomedical research with clinical practice represents an important and timely challenge. Overall, the proposed approach is sound and the presented results are convincing.

Impact:

- The study contributes new ideas and solutions to address a key question in current biomedical research and has thus potential to generate wide impact.
- The study could be of interest to clinical researchers and practitioners, to computer scientists working in the area of biomedical data integration, as well as to basic researchers working in a field related to any of the diverse data that go into the presented knowledge base, ranging from genetics to drug design. A prerequisite for this, however, is that the platform is made available to the community. I could not find detailed information whether this will be the case.

Strengths:

- The major strength of the study, in my view, lies in the very comprehensive and thorough data collection and integration. The knowledge base contains vast data from close to 30 databases, including gene-disease associations, phenotypic annotations, information on drugs and clinical data.
- The manuscript presents a range of relevant and well executed applications, substantiating the versatility of their platform.
- Even though the manuscript is rather dense on information, it is very well written and for the most part easy to follow.
- The 'clear box' paradigm is a clear advantage to previous machine learning approaches for finding interesting relationships in heterogenous biomedical datasets, as it allows for a relatively straightforward interpretation of its predictions.

- The study contains several individual results that I find particularly interesting. For example, the ability to dissect different kinds of relationship, e.g. BMI as a risk factor, symptom or side effect under different conditions (page 7). Or the observation that the learning rates, to some degree, recapitulates the discovery process of different relationships mentioned on page 11.

Weaknesses:

- The study presents an impressive proof of principle that large-scale biomedical knowledge can be integrated with electronic health records to provide accurate and meaningful predictions, annotations and interpretations. While I do not find this surprising in itself, I still appreciate the scale and thoroughness of the presented analyses.

- As a basic researcher, the most valuable aspect of the study for me personally would be to use the platform as a data resource and exploration tool. This would require (programmatic) access to the platform.

- I could not find any information on whether and how the platform will be made available to the community.

- While the manuscript sets out to advance the implementation of precision medicine, it mostly presents proof-of-concept validations of the general platform. More concrete applications to individual patient data would make a much stronger case for the potential of the platform in actual diagnosis / treatment / management / prognosis etc.

- As the predictions of the platform rely on the data fed into it, a more detailed presentation of the EHR data would be useful. For example, do the authors expect significant biases towards certain populations (age, sex, ethnicity etc.) and/or diseases?

- A major part of the knowledge base, the SPOKE platform, has been introduced before.

Methods:

- The study mostly uses well-established and reliable machine learning tools and concepts, such as heterogeneous networks and the PageRank algorithm. The major advance is thus perhaps not a methodological one, but lies in the careful application of established methods and the sheer scale of the integrated data.

- The different analyses are sound and well described.

Reviewer #3:

Remarks to the Author:

The authors propose a new method to integrate patient information from Electronic Health Records

(EHRs) to a heterogeneous knowledge network called Scalable Precision Medicine Oriented Knowledge Engine (SPOKE). That network includes relational information obtained from a set of publicly available datasets regarding anatomy, biological processes, cellular components, compounds, diseases, genes, etc. The main contribution of this work is to provide a way to determine the importance of each node in the SPOKE network regarding any code in the EHRs. The authors claim that this new methodology could help to uncover how different biomedical entities are related to each other, thus, speeding up the transformation between basic research and patient care. They also argue that the use of their approach could be helpful on the inference of new unobserved connections between diseases, diseases and genes, compounds, and compounds and genes.

Evaluation

The paper is well-written and, according to my opinion, their methodology could have a large impact on the understanding and discovery of relational characteristics of patient cohorts, since their descriptors could be easily applied as an input to advanced machine learning techniques. The idea of using a modified version of Page Rank to extract those descriptors is especially intuitive and interesting since it is based on a well-known metric of diffusion on complex networks. However, I have some major and minor concerns that should be clarified before I can recommend this work for publishing in Nature Communications.

Concerns:

- There is something that, after reading the document, it is not clear to me: the SPOKE network structure. According to Supplementary Table 1, there are different types of nodes and links, the question is: is that network structure a multilayer network? If it is, which are the rules that determine the random walk movements (there are some differences in RW on multilayer networks and monolayer networks)? If it is not, have the authors considered the impact of using multilayer structures?
- In results, authors state that: This immense enrichment occurred because, unlike the GWAS catalog 208 datasets in the Gene Expression Omnibus (GEO) with just BMI as a phenotype (without 209 any other major disease), had already been incorporated into SPOKE via obesity Disease210 UP(DOWN)REGULATES-Gene". We need to read the methods about how gene expression is part of Spoke. They should include some text in the results about the heterogeneous network.
- In the document, authors modify the standard PageRank algorithm weighting the re-start parameter of the random walker towards nodes that are important for a given patient population, but the definition about how they define this importance is not clear in the text. Since it seems that this modification is crucial, they should make an effort to better describe it.
- The authors use an interesting example to illustrate how their methodology works creating cohorts of patients based on their BMI, using a k-Means algorithm to perform the segmentation of patients, using some previous knowledge about the number of categories. However, there are many situations where the number of categories is unknown, and the usage of their methodology with the wrong number of cohorts could lead to spurious results. Did the authors check what the robustness of their approach in such scenarios?
- Which is the effect of the cohort size on the results that they report?
- On the discussion section, I missed some critical notes about the method. Even having some remarkable results, a note about the pitfalls and drawbacks of their methodology could provide a better understanding of the situations where their approach could be useful and the ones that could lead to undesired results. Besides, a critical analysis is mandatory according to the scientific method.
- Most of the figures present in the document must be redesigned.
 - o Figure 2:
 - The plots on the panel are not labeled.
 - The markers in (a) and (b) are not aligned with the labels of the x-axis, making them difficult to

read.

- Legends on (a) and (b) are too small and difficult to read.
- The colors of the lines in (a) are difficult to differentiate for individuals with color blind deficiency.

o Figure 3:

- Figure (a) is difficult to read, the legend is too small and the hierarchical structure is too dense.
- The authors do not mention with kind of linkage is used to perform the clustering.
- Maybe a different color palette, like viridis, could make the heatmap easy to read.

o Figure 5:

- There is an overlap between the different plots and subplots of this figure.
- Some of the circles on the networks are not closed.
- In (c), they do not specify, what the ribbon along the line means.

Reviewer 1

[1] Ln 94, Ln 103: MORE CLARITY ON PRESENTING KEY CONCEPTS WOULD BE GREAT! For readers without previous exposure to SPOKE I am afraid the result section's introduction of the central terms 'PSEV', 'SEP', 'HER concept' etc is rather a steep learning curve. It would facilitate understanding if the authors more explicitly describe the elementary (ontological) essence of these key elements, e.g. What kind of object (a vector!) is PSEV and of what (a concept??) is it a property? I know all the information is somehow implicitly there (e.g. V stands for vector) but rarely explicitly exposed and explained.

It sounds as if 'PSEV' represents (is a property of) a concept in the EHR (as repeated again in Ln 152), where a concept (in feature space) in the EHR can be a variable that is orthogonal (can be assigned) to any patient, such as BMI, a gene locus, a disease, a drug being taken by said patients... (please name examples upfront already in Ln 94 when explaining). Having this concrete examples would help greatly! Also please explicitly say that PSEV has the form of a vector that has as its N elements the weights assigned to each node in SPOKE. Also say what is N (the number of nodes in SPOKE?)

From the above explanation in the opening of the RESULT section, it also sounds as if the category to which the NODES in SPOKE belong and the category to which "CONCEPTS" in EHR (medical terms, that can be assigned to patients...] are one and the same and hence are allowed overlap? Is it fair to say that hence, the term 'CONCEPT' can refer to both, nodes in SPOKE and CONCEPTS sensu strictiore (medical variables...) in the EHR? This is the impression one gets.

We completely agree with the reviewer that key terms such as EHR concepts, SEPs, and PSEV should be clearly defined early in the manuscript. In order to address this comment, we have added the following sections:

- LN 91-97: These embeddings, called Propagated SPOKE Entry Vectors (PSEVs), can be created for any group of subjects with a particular characteristic (i.e. patient cohort). Here we describe the creation of PSEVs for patient cohorts selected using either discrete or continuous EHR variables. PSEVs are vectors in which each element corresponds to a node in SPOKE. Therefore, the length of each PSEV is equal to the number of nodes in SPOKE. Furthermore, the value of each element in a PSEV encodes the importance of its corresponding node in SPOKE for a given patient cohort.
- LN 105-110: Each structured EHR table contains codes, referred to as EHR concepts, that can be linked to standardized medical terminology. EHR concepts can be diagnostic codes (ICD9CM or ICD10CM), medication order codes (translated to RxNorm), or lab codes (LOINC). Any SPOKE node that can be directly linked to an EHR concept is a SPOKE Entry Point (SEP).

[2] But then there is a subtlety that becomes apparent from the first example and that questions the validity of the definition of PSEV in [1]: It turns out that the notion of mapping of EHR CONCEPTs to a PSEV is actually not precisely as defined in the text (Ln 94). The mapping is not between an EHR CONCEPT to PSEV, but between a VALUE (RANGE) within an EHR CONCEPT – one that defines a COHORT (a “natural” subpopulation). Thus, to be specific, ‘BMI’ is not the concept that maps into a PSEV but rather, a given value of BMI, i.e., what maps to a PSEV is not a CONCEPT (variable) but a particular instance (value) of a variable, such as a range of the BMI, e.g. Obesity Cohort IV.

This is CRUCIAL because this is where the biological meaning of a PSEV comes from: A PSEV is COHORT- SPECIFIC, where a cohort is defined by a specific natural property (patients having a particular disease X) WITH respect to a CONCEPT (disease X).

I think this relationship (mapping) should be emphasized and also indicated in the FIG 1, e.g. with a double-headed long arrow connecting the text “BMI Cohort IV” on the top with the text “Propagated SPOKE Entry vector” in the bottom. This would make a lot of things clearer.

To create a PSEV, the mapping is in fact from a specific EHR concept (e.g. obesity) to SPOKE, but you are correct in pointing out that a PSEV is cohort specific and we didn’t make that clear in the definition of a PSEV. Specifically, we use information in the EHRs at two stages in the creation of a PSEV. First (Fig 1B step 1), to identify all SPOKE Entry Points (SEPs). We do this by directly mapping EHR concepts (e.g. BMI, hypertension, acetaminophen, etc.) to nodes in SPOKE. Second (Fig 1B step 2), to define a cohort (e.g. BMI group IV). In this study we show that cohort selection can be carried out using either continuous variables (e.g. BMI, glycemia, etc.) or discrete variables (e.g. “multiple sclerosis”). We think the confusion comes from the fact that the EHR concepts used to create SEPs are also used for cohort selection. We chose to use SEPs for cohort selection because it allowed us to create high throughput benchmarks that could determine whether PSEV were able to recover deleted edges. In response to your concerns and suggestions, we have then decided to introduce the following changes:

- We have changed Fig 1B step 4 text to read “Propagated SPOKE Entry Vector for Cohort IV” to make it clear that PSEVs are cohort specific.
- Additionally, we have elaborated on the PSEV definition LN 95-98: “These embeddings, called Propagated SPOKE Entry Vectors (PSEVs), can be created for any group of subjects with a particular characteristic (i.e. patient cohort). In fact, PSEVs are vectors in which each element corresponds to a node in SPOKE. Here we describe the creation of PSEVs for patient cohorts selected using either discrete or continuous EHR variables.”
- We have also changed LN 239-244: “After identifying that the obese BMI PSEV was able to preserve the known gene expression edges in SPOKE, we decided to check this with other concepts in a high throughput manner. To do this, we utilized the fact that SEPs are SPOKE nodes that can be

directly mapped to EHR concept(s) to extract 3,233 patient cohorts from the diagnosis, medication order, and lab tables in the EHRs. PSEVs were then generated for each of the cohorts”

[3] Ln 105: The definition of SEPs is also semasiologically not kosher: “Each structured EHR table contains codes that can be linked to standardized medical terminology allowing direct links to SPOKE, referred to as SPOKE Entry Points (SEPs).” □ SEPs are NOT ‘links’ as this sentence implies; but an SEP, as I understand it, is a ‘node’ in SPOKE; more specifically it is a SUBSET of the set of all the nodes in SPOKE, defined by mapping to a concept that exists in the EHR. Is this correct?

You are absolutely correct. In order to address this, we have updated the definition of a SEP to:

- LN 108-110: “Any SPOKE node that can be directly linked to an EHR concept is a SPOKE Entry Point (SEP).”

Similarly fuzzy expression about the essence of SEPs is the statement Ln369:” PSEVs could find SEP to SPOKE level relationships”. Please avoid what philosophers call “category mistakes”.

Yes, that sentence makes it sound like SEPs aren’t nodes in SPOKE. That phrasing was a product of using EHR concepts (that were used to create SEPs) to select the cohorts. We completely understand why that would cause uncertainty about the definition of a SEP. As a result we have changed the sentence mentioned in LN 369 to the following:

- LN 394-498: “By correctly inferring the *Disease-Gene* and *Compound-Gene* edges, we demonstrated that PSEVs could uncover higher order relationships, such as those between a cohort and SPOKE. At the same time, correctly inferring *Disease-Disease* and *Compound-Compound* edges demonstrated that PSEVs could uncover relationships among EHR concepts themselves.”

[4] The FIGURE 1B offers a much needed help for understanding the concepts but it could be improved in some points:

-4a- Please indicate (label) with an arrow pointing to a specific element in the graph, what is the actual entry points (SEP), namely: the colored circles

Please see new hypertension example near step 1

-4b- Also use a label/arrow to indicate that the entire horizontal bar at the bottom (step 4) is the PSEV.

Added label

-4c- With another label/arrow describe what a small square in that PSEV bar represents: Do they represent ALL the nodes of SPOKE (each square = one node), with the white ones representing nodes that are NOT SEPs and hence have no weight (weight = 0)? (Just my educated guess). It would help to repeat that the PSEV is essentially a weighted vector that lists all the nodes in SPOKE (if that is the case).

Thank you for the suggestion. Please see bottom of Figure 1B

-4d- Another label/arrow is needed that explains that each of the green empty square in Step 1 represents a patient (I believe) of that one given cohort.

Added label

-4e- It may help to show in the box “UCSF EHRs” a simple cartoon of an entire EHR table with patients as rows and CONCEPTS has column heading, of which one is BMI, and label the columns as “CONCEPT and connect one of them to one of the colored circle in the SEP vector of Step 1.

Added cartoon table

-4f- The label ‘SEPS’ should be ‘SEPs’ (plural of SEP)

Fixed

[5] Ln 126: Not sure why k-means is used to generate the four classes of BMI. It makes sense to not use “a priori” defined classes which I guess are defined by numerical ranges of BMI and not shown in the graph. By contrast the authors observed four “peaks” (subpopulations) in their 800K UCSF patients, shown in Fig 1A. Are these the same as the “a priori defined”? Why not used four natural subpopulations? The fact that the k-means group boundaries do not coincide with the four clearly distinguishable natural subpopulations in the multi-modal distribution of the UCSF cohort means that the standard deviations of the individual subpopulations is not invariant.

You are correct in that we could easily use the natural boundaries shown in the distribution. In fact, BMI and its standard classification (underweight, normal, overweight, and obese) are very relatable. We believe that by staying consistent with the idea of four BMI classes will make it easier for the reader to follow along with paper. The reason we originally chose to use K-means (n=4) was to keep our threshold choices as un-biased as possible, we chose to use k-means (k=4). That being said, we introduced a number of changes and additions to address your point as follows:

- We now use the natural boundaries instead of K-means.
- We added an additional analysis that shows it is unnecessary to use classes at all. Supplementary Figure 2 shows that the trends observed when separating patients into BMI classes is also observed when treating BMI as a continuous variable.
- We added the following text to the manuscript LN208-211: “Further, these results were replicated treating BMI as a continuous variable instead of discrete classes (Supplementary Figure 2). This replication shows the robustness of this approach and is important given that most continuous variables in the EHRs are not associated with a fixed number of classes.”

I am not sure if by “a priori” the authors mean (i) these four peaks from the data or (ii) some numerical ranges defined elsewhere. In any case, the current description in the text is not logically coherent but could readily be corrected by saying what is meant by “a priori”.

If correcting this point, please also modify the text at Ln186 to make it consistent.

- Thank you. In LN 140-142 we now write “Therefore, patient cohorts can be created without a priori knowledge of any standard or pre-defined classes.”

[6] Ln131: The PageRank algorithms is actually not well explained. The text points to FIG 1B which does not provide a good explanation, and the legend is very cursory in this respect. What does (Ln 136) “jump back to the patient population with probability beta” mean? I think one can only jump back to a node, not to a ‘population’. (This is one of many mild category mistakes in this ms). Do you mean jump back to a node that is an SEP which is linked to a concept in the specific COHORT (hence a ‘population’)? In the Methods on p.17, Ln459 there is a differing –perhaps better– explanation for what is the probability of beta: “ ...for jumping out of SPOKE to the given SEP”. This sounds more precise but what is a “given SEP”? Is it what I suspect in the previous sentence? What does “out of SPOKE” mean? –Do you mean the SPOKE nodes that do NOT represent SEPs??

But in Ln459 in the METHODS, beta is defined as the “probability for a random jump”. (ANY jump?) So what is beta?

It is also difficult to relate beta to anything in the original PageRank and the reference to the original papers are not very helpful. Is the random walk the equivalent to the “random surfer” and is (beta-1) sort of like the damping factor d in the original PageRank algorithm?

Thank you for the insightful comments and suggestions. In the original PageRank paper, section 6 Personalized PageRank the authors talk about the vector E that allows the random surfer to avoid sinks (such as cycles with no outgoing edges) by giving the surfer the ability to jump randomly to any node in the network. Usually E is uniform (β / N where β = the probability of random jump (restart parameter or 1-damping factor) and N = number of nodes in the network). As the authors explain, though this technique has been very successful, it is possible to customize the ranks of nodes by adjusting their values with E .

In our case E is adjusted in two ways before creating each PSEV. The first change is that when the surfer randomly jumps, it must jump to a node that is a SEP (not to any node in SPOKE). Second, the probability of jumping to SEP_i is based on the proportion of patients (in the cohort of patients being used to create PSEV $_j$) that had SEP_i in their records. This value is considered to be a measure of the importance of that node for the cohort in question, and thus a higher probability of return to that node is allowed.

We now realize that in our original manuscript we didn’t thoroughly describe the way in which the algorithm was adapted (particularly the changes to the vector E).

- To address this shortcoming, we now added Supplementary Figure 1 and amended the text (LN 147-154) as follows: “Third, a random walker was initialized and allowed to either move to a neighboring node (optimized damping factor=0.9) or randomly jump to any SEP with probability β (optimized $\beta=0.1$). However, β was not evenly distributed among the SEPs (as in the original algorithm), but was instead weighted based on how important each SEP was for the cohort (Supplementary Figure 1). This weight is akin to having the random walker jumping to a random patient in the cohort and traversing to one of that patient’s SEPs (Supplementary Figure 1A).”

[7] Another related unclear explanation is the term ‘rank vector’ that results from the random walk (Ln137). Is this essentially the PSEV? Or how do you get to the PSEV from here?

In the original PageRank paper, section 2.6 describes the computation of PageRank. It explains that algorithm iterates until it converges. The idea is that if the rank vector at iteration i is similar enough to the rank vector at $i-1$ then it has converged. Once the algorithm converges we call the rank vector a PSEV. We address this in the text by writing

- LN 157-159, “Once the algorithm converged, the rank vector from the final iteration was returned (bottom vector). This final rank vector is called a PSEV.”

Until now, the reader is left with the impression (through guessing) that the PSEV is a vector that contains N elements (for the N nodes in the SPOKE) with the values representing some “weight” indicating the importance of a node-concept in SPOKE for THE EHR-concept (disease X of a cohort) to which the PSEV is associated to. This is important for later when “ranks” of diseases are compared for the various PSEVs of the BMI categories (FIGURE 2A). Please articulate all these relationships more explicitly and clearly. This relates to my points [1] and [2].

You are absolutely correct. We now believe that through the multiple changes introduced in response to your (and the other reviewers’) questions/concerns, the reader will get a clearer impression of our method.

[8] Ln 152: It will help for understanding if the authors take an opportunity to restate the definition of PSEV by modifying the sentence here to (if I am correct): “The final PSEV is representative of how important EACH node of SPOKE is, not just the subset of nodes that comprise the SEPs.”

This would also confirm that the number N of components of the vector PSEV vector is the same as the number of nodes N in SPOKE. The reader had to assume this all along since it is not explicitly articulated (although inferable).

It may also help here to restate that each BMI category (I, II, III, IV) has its own PSEV – this would help the comprehension of FIG 2A (since so far PSEV has been, perhaps sloppily, equated to a EHR concept, and not a subclass within a concept (defined by a cohort), as discussed above, point [2])

You are correct. This comment about the length of the PSEV has now been addressed throughout the text. We have also added the following sentence:

- LN 171-173, “Therefore, a PSEV was generated for each of the four BMI cohorts. Since each element in a PSEV corresponds to a single node in SPOKE it is now possible to determine how important each SPOKE node is for each of the BMI cohorts.”

[9] Ln163 on: What is “average BMI?” The FIGURE 2A (which is hard to read) relates to the 4 classes of BMI. Why not say here: “To learn more about the relationships between the four categories of BMI (I=Underweight, II=normal weight, III=Overweight, IV=Obese) and these potentially associated diseases, we plotted for each of these categories the rank of these Disease elements in the four respective PSEVs (Figure 2A)”. Then please improve FIGURE 2A:

Average BMI is the average BMI in a BMI cohort. Since each cohort has a range of BMIs we decided to use the mean.

We changed the manuscript to say:

- LN 183-189: “To learn more about the relationships between BMI and these potentially associated diseases, for each BMI class, we plotted the average BMI (mean BMI per cohort) against the rank of these *Disease* elements in the four respective PSEVs (Figure 2A).”

-9a- Add the labels I, II, III, IV to the X-axis categories. It is confusing that the authors suddenly switch to the names of these categories without mentioning the equivalence

Fixed

-9b- The legend of INSET (I guess the diseases chosen) is too small – UNREADABLE!

Fixed

-9c- The Y-axis should say “Rank of Disease in the PSEVs”

Fixed

[10] Ln193 etc: The sloppily metaphoric verbiage, such as “hypertension MOVES...” or a disease undergo “Rank change” is confusing. I assume all this is related to moving from lower to higher BMI categories. Please say so explicitly – the reader has not internalized the framework that the authors have in mind when they use the terminology of “movements” in this purely static analysis

We very much agree with this comment and have replaced the metaphoric text with the following:

- LN 186-189: “The most noticeable differences in the rank were observed for the *Disease* element hypertension between the underweight and normal cohorts (rank increases from 136 to 17) and the *Disease* element obesity between overweight and obese (rank increases from 132 to 1).”
- LN 192-193: “Additionally, there was a major difference (111 positions) in the rank of the type 2 diabetes mellitus between the normal and overweight cohorts.”

[11] Ln192 ff: For the discussion of association with GENOTYPE – the terminology has to be more precise. What does it mean “Genes that are positively correlated with average BMI”. I assume every gene discussed is present in everybody. Also what does “average” BMI means? Within a category?

To make meaning of average BMI clear we added:

- LN 223-225: “Again, the genes that positively correlated with average BMI (mean BMI of a cohort) were given the top prioritization in the high BMI (obese) PSEV.”

What is a GENE?? Shouldn't it say, instead: “gene LOCI for which there is a variant (SNP) that is associated with BMI...”?

You are correct. The SNP -> gene problem is pervasive in all GWAS, as naturally, the associations are described between a given SNP and the phenotype. While an elaborated discussion of this problem is outside of the scope of the present work, we simply relied on the GWAS catalog for the genomic annotation for each SNP. The GWAS catalog uses the Ensembl mapping pipeline for their genomic annotations. So here we are essentially saying the top ranked *Gene* elements in the obese BMI PSEV have a higher probability of harboring a susceptibility variant.

It should be noted that repeating the BMI analysis using only exonic SNPs still resulted in significant enrichment of high BMI associated genes in the top ranked *Gene* elements in the obese BMI psev ($p=1.7e-7$).

We now address this issue in the following:

- LN 234-235: “Therefore, *Gene* elements that were highly ranked in the overweight or obese BMI PSEVs had a higher probability of harboring a susceptibility variant.”

[12] Ln205: Moreover, this same section, entitled “...Learn Genotypic Traits..”, suddenly sways off the topic and discusses ALTERED GENE EXPRESSION. The concept of gene expression is part of the phenotype and is not necessarily directly linked to genotype.

We agree that the sudden switch to the discussion of gene expression results is off topic. We have decided to remove the gene expression results from the manuscript.

[13] Ln 221: “If the top 39...” should start a new sentence.

Fixed

[14] Ln 255: In moving towards the heatmap, the authors utilize the concept of the modality of the edges in SPOKE, but have not explicitly introduced it in this ms. other than mentioning the number of 24 TYPES of edges in the INTRODUCTION. As said, I would not rely on readers being already familiar with SPOKE. Somewhere, perhaps here, the nomenclature and essence of edges should be briefly defined. How is a Disease-Disease or a Disease-Gene link established in SPOKE? (Give example)

Thank you for bringing this to our attention. To address this comment we have now added the following text to the manuscript:

- LN 73-77 “Currently, SPOKE integrates data from 29 publicly available databases, such as the GWAS catalog, STARGEO, ChEMBL, LINCS, and GeneOntology, and contains over 47,000 nodes of 11 types and 2.25 million edges of 24 types, including disease-gene, drug-target, drug-disease, protein-protein, and drug-side effect^{11,12}.”
- LN 246-252: “For example, the Multiple Sclerosis (MS) *Disease* node is connected to 39 *Anatomy* nodes (such as *MS-LOCALIZES_DIA-Central Nervous System*) in SPOKE. Notably, there is 100% overlap between the top 39 ranked *Anatomy* elements in the MS PSEV and all actual MS *Anatomy* neighbors (n=39). Similarly, for *Symptom* nodes connected to MS (such as *MS-PRESENTS_DpS-Fatigue*), 80% of first neighbor relationships are maintained in the top n- *Symptom* elements of the MS PSEV.”
- LN 272-277: “Additionally, a series of benchmarks (supplemental text) shows that PSEVs ability to learn connections can be applied to other edge types such as *Disease-Disease* (edges from MEDLINE co-occurrence) and *Compound-Compound* similarity (edges DICE similarity), *Compound* to drug-protein (molecular targets; edges from DrugBank, DrugCentral, BindingDB), and *SideEffect-Anatomy* (edges from MEDLINE co-occurrence).”
- LN 284-288: “To address this point, all of the *Disease-Disease* (e.g. *MS-RESEMBLES_DrD-Amyotrophic Lateral Sclerosis*) and *Disease-Gene* edges (*MS-ASSOCIATES_DaG-IL7R* and *MS-DOWNREGULATES-PALLD*) in SPOKE were removed and the PSEVs were recomputed the *Disease* PSEVs (PSEV^{ADD, ΔDG}).”

What does “ASSOCIATES-DaG” (Ln273) and what does “REGULATES” (Ln303) mean exactly?

ASSOCIATES-DaG refers to genes that are influential or genetically associated (GWAS Catalog, DisGeNET, DISEASES, or DOAF), while REGULATES refer to gene expression profiling (from StarGEO). We added the following:

- LN 284-288: “To address this point, all of the *Disease-Disease* (e.g. *MS-RESEMBLES_DrD-Amyotrophic Lateral Sclerosis*) and *Disease-Gene* edges (*MS-ASSOCIATES_DaG-IL7R* and *MS-DOWNREGULATES-PALLD*) in SPOKE were removed and the PSEVs were recomputed (PSEV^{ADD, ΔDG}).”

- LN 304-306: “In SPOKE, the ASSOCIATES_DaG edges represent known associations between *Diseases* and *Genes* and were obtained from the GWAS Catalog¹⁵, DISEASES¹⁶, DisGeNET^{17,18}, and DOAF¹⁹.”
- LN 331-334: “We found that the PSEVs learned well-established (ASSOCIATES from the GWAS Catalog, DisGeNET, DISEASES, or DOAF) *Disease-Gene* edges before the noisier (UP(DOWN)REGULATES from StarGEO) edges (Figure 4B).”

[15] FIGURE 3A: More details are needed here too. How the heatmap was constructed should be explained in more detail. It is essentially a bi-clustering of a data in an M x N table, eg with rows = diseases and columns = genes. But how were they selected, or extracted, from the PSEVs?? The Legends is better than the text but still not sufficient. Also please label the two axes (ROWS vs COLUMNS) of the heat map with: ‘GENES’, ‘DISEASES’ (this always helps even if it is in the legend)

The heatmap is showing *Gene* ranks (columns) within each of the 137 *Disease* PSEVs (rows). Only *Genes* that have at least 1 *Disease-Associates-Gene* edge in SPOKE were used.

- We added labels to rows and columns.
- Added to the caption of Figure 3: “Heatmap generated with the *Disease* PSEV^{ADD, ΔDG} (only using elements of *Genes* that associate with at least one *Disease*). The heatmap shows the *Gene* ranks (columns) within each of the 137 *Disease* PSEVs (rows).”

[16] Ln303: Why is the edge “REGULATES” (whatever it means, see comment above) more “noisy” than “ASSOCIATES” given that the former is a molecular interaction and the latter just a cohort observation?

The REGULATES edges come from StarGEO while the ASSOCIATES edges come from the GWAS catalog. Generally, genetic associations are identified through large studies (with thousands of subjects) and unequivocally replicated in independent populations. In contrast, gene expression profiling experiments are usually reported by single labs, with no replication, and small n. This causes the latter to be less reliable (thus noisier) than the former.

[17] DISCUSSION. While I am empathetic to the author’s enthusiasms about the potential utility of their platform, and prefer short discussions (as it is now), I think that one paragraph on (current) limitations leading to an outlook (where precisely one could improve) would be in order.

The reviewer raises an interesting point. We have now added the following reflection to the manuscript:

- LN 412-430: “The main limitations of this approach mostly stem from the potential inaccuracies in the EHRs and the incompleteness of the knowledge networks (SPOKE). First, while maintaining the trust and privacy of patients remains paramount, it has also made it difficult for institutions to share even de-identified records. Not being able to openly share data means that the patient population used may not be representative of the general population, especially in terms of race, ethnicity, education and income. Second, many institutions don’t use standardized medical terminology, thus making it challenging to accurately map EHR concepts to SPOKE. That being said, institutions that use EHR formats that utilize standard terminologies, like the Observational Medical Outcomes Partnership (OMOP) Common Data Model, can easily implement this in their own system. While we did not use OMOP in this work, efforts by our group and others are ongoing in this direction. Finally, we are limited by the fact that as long as our biomedical knowledge is incomplete the same will be true for our knowledge networks. In this regard, SPOKE is continually under development and future versions will increase in complexity and completeness. However, our results show that adding context with the EHRs actually enabled us to learn new relationship in the network, thereby growing our biomedical knowledge. We believe that these limitations are inherent to this field of study and that the development of tools, such as the one presented here, can spur collaboration between institutions and help overcome these limitations. “

Reviewer 2:

The study presents an impressive proof of principle that large-scale biomedical knowledge can be integrated with electronic health records to provide accurate and meaningful predictions, annotations and interpretations. While I do not find this surprising in itself, I still appreciate the scale and thoroughness of the presented analyses.

We thank the reviewer for his/her comment on our work.

As a basic researcher, the most valuable aspect of the study for me personally would be to use the platform as a data resource and exploration tool. This would require (programmatic) access to the platform.

I could not find any information on whether and how the platform will be made available to the community.

As required for Nature Communications, our code will be made available (on GitHub) to the public. Additionally, to make our results available to those without a programming background, we developed a platform where users can browse and download the top 100 ranked nodes (of each node type) for over 1,000 diagnosis or medication EHR concepts. To use this platform please visit:

PSEVexplorer.ucsf.edu

While the manuscript sets out to advance the implementation of precision medicine, it mostly presents proof-of-concept validations of the general platform. More concrete applications to individual patient data would make a much stronger case for the potential of the platform in actual diagnosis / treatment / management / prognosis etc.

You are absolutely right. While specific examples on the utilization of this approach are underway, we deemed necessary to describe our methodology in detail in this first publication. Thus the aim of this study, as you point out, is simply to present “an impressive proof of principle that large-scale biomedical knowledge can be integrated with electronic health records to provide accurate and meaningful predictions, annotations and interpretations”. That being said, we have developed more concrete applications to individual patients applications. However, we thought it was important to layout the proof of principal so it can be peer reviewed and presented to the community before we publish the downstream applications.

As the predictions of the platform rely on the data fed into it, a more detailed presentation of the EHR data would be useful. For example, do the authors expect significant biases towards certain populations (age, sex, ethnicity etc.) and/or diseases?

In many ways, yes, we do expect to observe significant bias, the most obvious of which is race as the patient population at UCSF doesn't reflect that of the general population. In our opinion, the only way to get around this issue is by collecting EHRs from other hospitals throughout the country (and the world) therefore increasing the racial diversity of patients. Though the sharing of EHRs between hospitals can currently be very challenging, we hope that the development new

methods, like the one presented here, will encourage the sharing medical information so that we can make tools that are fair and unbiased. We now elaborate on this limitation of our approach in the Discussion.

A major part of the knowledge base, the SPOKE platform, has been introduced before.

This is correct. As such, we generally refer the interested reader to our previous paper. However, we deemed necessary to reiterate some basic concepts here to help the reader interpreting a fairly complicated and unconventional method.

Reviewer 3

There is something that, after reading the document, it is not clear to me: the SPOKE network structure. According to Supplementary Table 1, there are different types of nodes and links, the question is: is that network structure a multilayer network? If it is, which are the rules that determine the random walk movements (there are some differences in RW on multilayer networks and monolayer networks)? If it is not, have the authors considered the impact of using multilayer structures?

We have now tried to clarify the structure of the network along several passages in the manuscript. To specifically answer your question, SPOKE is not currently a multi-layer network. All our efforts went into integrating as many databases as possible and treated the resulting ensemble as a heterogeneous network. However, the idea of exploring SPOKE as a multi-layered network is intriguing and we will consider it in the future.

In results, authors state that: This immense enrichment occurred because, unlike the GWAS catalog 208 datasets in the Gene Expression Omnibus (GEO) with just BMI as a phenotype (without 209 any other major disease), had already been incorporated into SPOKE via obesity Disease210 UP(DOWN)REGULATES-Gene". We need to read the methods about how gene expression is part of Spoke. They should include some text in the results about the heterogeneous network.

Due a comment from Reviewer #1, that the discussion of gene expression in this section "suddenly sways off the topic", we have decided to remove this part of the analysis. However, we completely agree with you that the results needed more text about SPOKE. We addressed this by adding information about the sources of SPOKE edges as well as examples of actual edges. The specific edges you refer to here were address in the following:

- LN 284-288: "To address this point, all of the *Disease-Disease* (e.g. *MS-RESEMBLES_DrD-Amyotrophic Lateral Sclerosis*) and *Disease-Gene* edges (*MS-ASSOCIATES_DaG-IL7R* and *MS-DOWNREGULATES-PALLD*) in SPOKE were removed and the PSEVs were recomputed ($PSEV^{\Delta DD, \Delta DG}$).
- LN 304-306: "In SPOKE, the ASSOCIATES_DaG edges represent known associations between *Diseases* and *Genes* and were obtained from the GWAS Catalog¹⁵, DISEASES¹⁶, DisGeNET^{17,18}, and DOAF¹⁹."
- LN 331-334: "We found that the PSEVs learned well-established (ASSOCIATES from the GWAS Catalog, DisGeNET, DISEASES, or DOAF) *Disease-Gene* edges before the noisier (UP(DOWN)REGULATES from StarGEO) edges (Figure 4B)."

In the document, authors modify the standard PageRank algorithm weighting the re-start parameter of the random walker towards nodes that are important for a given patient population, but the definition about how they define this importance is not clear in the text. Since it seems that this modification is crucial, they should make an effort to better describe it.

Thank you for bringing up this shortcoming, which Reviewer #1 also identified. In the original PageRank paper, section 6 Personalized PageRank the authors talk about the vector E that allows the random surfer to avoid sinks (i.e. cycles with no outgoing edges) by giving the surfer the ability to jump randomly to any node in the network. Usually E is uniform (β / N where β = the probability of random jump (restart parameter or 1-damping factor) and N = number of nodes in the network). As the authors explain, though this technique has been very successful, it possible to make the ranks of nodes more personalized by adjusting the values with E .

In our case E is adjusted in two ways before creating each PSEV. The first change is that when the surfer randomly jumps they must jump to a node that is a SEP (not to any node in SPOKE). Second, the probability of jumping to SEP_i is based on the proportion of patients (in the cohort of patients being used to create PSEV_j) that had SEP_i in their records.

We now realize that the original manuscript didn't thoroughly describe the way we adapted the algorithm (particularly the changes to the vector E). To address this we added Supplementary Figure 1 and amended the text as follows:

- LN147-154: "Third, a random walker was initialized and allowed to either move to a neighboring node (optimized damping factor=0.9) or randomly jump to any SEP with probability β (optimized $\beta=0.1$). However, β was not evenly distributed among the SEPs (as in the original algorithm), but was instead weighted based on how important each SEP was for the cohort (Supplementary Figure 1). This weight is akin to having the random walker jumping to a random patient in the cohort and traversing to one of that patient's SEPs (Supplementary Figure 1A)."

The authors use an interesting example to illustrate how their methodology works creating cohorts of patients based on their BMI, using a k-Means algorithm to perform the segmentation of patients, using some previous knowledge about the number of categories. However, there are many situations where the number of categories is unknown, and the usage of their methodology with the wrong number of cohorts could lead to spurious results. Did the authors check what the robustness of their approach in such scenarios?

Thank you for your comment and suggestion. We are no longer using k-Means due to a comment from Reviewer #1, that we should use the natural sub-population instead of k-Means. That being said, it is very true that in most cases, when studying continuous variables, the number of categories is unknown (and there might not be such obvious sub-populations). To address this issue, we ran

the same analysis, but instead of using categories (natural sub-populations or k-means) we treated BMI as a continuous variable (Supplementary Figures 2) and observed the same results. We added the following text to the manuscript:

- LN208-211: “Further, these results were replicated treating BMI as a continuous variable instead of discrete classes (Supplementary Figure 2). This replication shows the robustness of this approach and is important given that most continuous variables in the EHRs are not associated with a fixed number of classes.”

Which is the effect of the cohort size on the results that they report?

This is an excellent observation. Cohort size can significantly impact the PSEV. In general, the effect on size really depends on the complexity of the cohort. If the patient cohort is comprised of patients with common symptoms, medications, and lab tests then only a few patients will suffice to create a robust PSEV. However, a complex cohort is under consideration, with multiple therapeutic interventions, and significant disease heterogeneity, then more patients will be needed. To address this point, in the supplemental text we compare the generated PSEVs vs three different types of random PSEVs. The extent to which the PSEVs differ from their random counterparts provides an idea as to whether the size of the cohort matched the complexity of the cohort.

On the discussion section, I missed some critical notes about the method. Even having some remarkable results, a note about the pitfalls and drawbacks of their methodology could provide a better understanding of the situations where their approach could be useful and the ones that could lead to undesired results. Besides, a critical analysis is mandatory according to the scientific method.

We thank the reviewer for bringing this important point. We have now evaluated some of the most relevant limitations of our approach and added the following text:

- LN 412-430: “The main limitations of this approach mostly stem from the potential inaccuracies in the EHRs and the incompleteness of the knowledge networks (SPOKE). First, while maintaining the trust and privacy of patients remains paramount, it has also made it difficult for institutions to share even de-identified records. Not being able to openly share data means that the patient population used may not be representative of the general population, especially in terms of race, ethnicity, education and income. Second, many institutions don’t use standardized medical terminology, thus making it challenging to accurately map EHR concepts to SPOKE. That being said, institutions that use EHR formats that utilize standard terminologies, like the Observational Medical Outcomes Partnership (OMOP) Common Data Model, can easily implement this in their own system. While we did not use OMOP in this work, efforts by our group and others are ongoing in this direction. Finally, we are limited by the fact that as long as our

biomedical knowledge is incomplete the same will be true for our knowledge networks. In this regard, SPOKE is continually under development and future versions will increase in complexity and completeness. However, our results show that adding context with the EHRs actually enabled us to learn new relationship in the network, thereby growing our biomedical knowledge. We believe that these limitations are inherent to this field of study and that the development of tools, such as the one presented here, can spur collaboration between institutions and help overcome these limitations.”

- Most of the figures present in the document must be redesigned.

o Figure 2:

- The plots on the panel are not labeled.

Fixed

- The markers in (a) and (b) are not aligned with the labels of the x-axis, making them difficult to read.

Fixed

- Legends on (a) and (b) are too small and difficult to read.

Fixed

- The colors of the lines in (a) are difficult to differentiate for individuals with color blind deficiency.

Fixed

o Figure 3:

- Figure (a) is difficult to read, the legend is too small and the hierarchical structure is too dense.

Fixed

- The authors do not mention with kind of linkage is used to perform the clustering.

Within the original methods we state that we used “seaborn clustermap package in python with the settings method='average' and metric='euclidean'.” However, to address your comment we have added: “Here, method refers to the method used to calculate the linkage and metric is the way in which we calculate the distance within the method.”

- Maybe a different color palette, like viridis, could make the heatmap easy to read.

We have eliminated some clutter and used a bigger font, thus making the figure more readable.

o Figure 5:

- There is an overlap between the different plots and subplots of this figure.

Fixed

- Some of the circles on the networks are not closed.

Fixed

- In (c), they do not specify, what the ribbon along the line means.

To address this comment we added to the caption:

- Ribbon in (C) shows range of fold change for different values of β (plots A-B use optimized $\beta=0.1$).

Reviewers' Comments:

Reviewer #1:

Remarks to the Author:

Nelson et al. have now provided a carefully revised manuscript in which they have addressed all the concerns I had raised. They have dealt with the challenges of conveying a complicated work such that the ms. reads very well. Especially the figures exhibit much didactic sensibility and a very clear.

As said before this is an important piece of work of potentially great utility and a first in its class. But even after rereading, I find that it will be challenging to fully comprehend all the new concepts introduced in it unless one actually uses the system -which I hope will soon gain a large user community that will test its robustness in a wider range of applications.

I have few remaining minor points (within the revised sections):

(1) Lines 105-110: The term 'SEP' is used in line 105 without explanation and prior mention; it is only explained further below, including introduction of the acronym, on line 110.

(2) Lines 121: The sudden introduction of a suffix for SEP (SEP_i) is confusing. May be clearer (I am guessing about the intention), if the authors would say: "Hence, the importance of a GIVEN SEP_i is equivalent to the proportion of patients in the cohort that had an EHR concept IN THEIR EHRs that mapped INTO THAT SEP_i." (?)

(3) Line 141: How do you "create" a patient cohort? Rather: you create a PSEV for a given cohort defined by a clinical characteristic of interest.

Reviewer #2:

Remarks to the Author:

The revised version has considerably improved, all my comments have been addressed.

Reviewer #3:

Remarks to the Author:

In the revised version of this work, the authors have tackled most of my concerns regarding their scientific claims and the structure of the article. I think that the manuscript has increased their readability and their scientific claims are more clear and argued. Their results could have a strong impact on the scientific community. As a result, I strongly recommend its publication on Nature Communications.

REVIEWERS' COMMENTS:

Reviewer #1 (Remarks to the Author):

Nelson et al. have now provided a carefully revised manuscript in which they have addressed all the concerns I had raised. They have dealt with the challenges of conveying a complicated work such that the ms. reads very well. Especially the figures exhibit much didactic sensibility and a very clear.

We thank this reviewer for the positive feedback!

As said before this is an important piece of work of potentially great utility and a first in its class. But even after rereading, I find that it will be challenging to fully comprehend all the new concepts introduced in it unless one actually uses the system -which I hope will soon gain a large user community that will test its robustness in a wider range of applications.

I have few remaining minor points (within the revised sections):

(1) Lines 105-110: The term 'SEP' is used in line 105 without explanation and prior mention; it is only explained further below, including introduction of the acronym, on line 110.

We have addressed this by defining SEPs earlier in the manuscript:

- LN 107: These overlapping concepts between the EHRs and SPOKE are called SPOKE Entry Points (SEPs).

(2) Lines 121: The sudden introduction of a suffix for SEP (SEP_i) is confusing. May be clearer (I am guessing about the intention), if the authors would say: "Hence, the importance of a GIVEN SEP_i is equivalent to the proportion of patients in the cohort that had an EHR concept IN THEIR EHRs that mapped INTO THAT SEP_i." (?)

We have taken the review's suggestion and edited this sentence:

- LN 124-126: Hence, the importance of a given SEP (SEP_i) is equivalent to the proportion of patients in the cohort that had an EHR concept in their records that mapped to SEP_i

(3) Line 141: How do you "create" a patient cohort? Rather: you create a PSEV for a given cohort defined by a clinical characteristic of interest.

Reviewer is correct. We now changed this line as follows:

- LN 146-147: Therefore, a priori knowledge about a cohort is not necessary to create a meaningful PSEV.